# Alongshore sediment transport analysis for a semi-enclosed basin: a case study of the Gulf of Riga, the Baltic Sea

Tarmo Soomere[1,2], Mikolaj Zbigniew Jankowski[1], Maris Eelsalu[1], Kevin Ellis Parnell[1], Maija Viška[3]

[1]Department of Cybernetics, School of Science, Tallinn University of Technology, Tallinn, 19086 Estonia
[2]Estonian Academy of Sciences, Kohtu 6, Tallinn, 10130 Estonia
[3]Latvian Institute of Aquatic Ecology, Rīga, LV-1007 Latvia

*Correspondence to*: Tarmo Soomere (tarmo.soomere@taltech.ee)

**Abstract.** The properties of wave-driven sediment transport and the dimensions of single sedimentary compartments are often radically different in different parts of semi-enclosed water bodies with an anisotropic wind climate. The western,
southern and eastern shores of the Gulf of Riga are a remote part of the more than 700 km long interconnected sedimentary coastal system of the eastern Baltic Sea from Samland in Kaliningrad District, Russia, to Pärnu Bay, Estonia. Even though shores of the gulf are generally straight or gently curved, the presence of small headlands and variations in the orientation of the coastline give rise to numerous fully or partially separated sedimentary compartments. We decompose sedimentary shores of this gulf into single compartments and cells based on the analysis of wave-driven potential sediment transport
using high-resolution wave time series and the Coastal Engineering Research Centre (CERC) approach. The western shore of the Gulf of Riga forms a large interconnected sedimentary system with intense sediment transport that is largely fed by sand transported from the Baltic proper. The southern shore has much less intense sediment transport and mostly accumulation areas. The south-eastern sector of the gulf is an end station of counter-clockwise sand transport. The eastern shore consists of several almost isolated sedimentary cells and contains a longer segment where clockwise transport
predominates. The transport rates along different shore segments show extensive interannual variations but no explicit trends in the period 1990–2022.

## 1 Introduction

Wave-driven sediment transport in the surf zone is a core process that shapes the shores of seas and oceans, including the key drivers of beach profile change, functioning of the cut and fill cycle, and loss of sediment to the offshore via driving surf
zone turbulence (Aagaard et al., 2021). It is also the principal agent of coastal erosion, alongshore sediment transport and sediment accumulation in the vicinity of the shoreline. These processes can be unidirectional, or circulatory on comparatively straight open ocean shores where waves usually approach the shore from a specific direction or at a small angle and where major headlands commonly divide the sedimentary system into large cells and extensive compartments (Thom et al., 2018).

The situation is complicated in water bodies of complex shape, such as the Baltic Sea (Fig. 1) where waves often approach the shore at large angles (Soomere and Viška, 2014; Eelsalu et al., 2024a; Soomere et al., 2024). The interplay of a high angle of approach and wind patterns with multi-peak directional structure gives rise to exceptionally powerful alongshore sediment flux (Viška and Soomere, 2013b) under a fairly modest wave climate (Björkqvist et al., 2018; Giudici et al., 2023), specific mechanisms that stabilise almost equilibrium beaches (Eelsalu et al., 2022), and persistent sediment

flux divergence areas which are most likely erosion hotspots at certain locations with small changes in the orientation of the coastline (Soomere and Viška, 2014; Eelsalu et al., 2023). To better characterise such situations, we use the term "cell" to denote relatively small coastal segments, elementary sedimentary units that are either mostly separated from the neighbouring segments or exhibit other clearly identifiable features (e.g., cells of predominantly one-directional sediment transit versus cells with almost no net sediment transport). In a similar manner, we use the term "compartment" to denote

clusters of cells that usually exchange sediment within the cluster but have either very limited sediment exchange with other compartments (e.g., because of the presence of a major divergence area of sediment flux), or only one-way sediment exchange with a neighbour.

Massive alongshore sediment transport is one of the main reasons for extensive coastal erosion (Eberhards et al., 2009) and the formation of large accumulation features that sometimes occur at a large distance from the erosion areas in the

eastern Baltic Sea (Tõnisson et al., 2016). This transport may amplify the impact of coastal defence structures on sediment deficit (Bernatchez and Fraser, 2012). It can also be a major problem from the viewpoint of coastal infrastructure design and maintenance (Bulleri and Chapman, 2010), the management of urban coastal landscape and for increasing resilience of coastal socio-ecological systems (Villasante et al., 2023). Persistent sediment flux divergence areas may serve not only as erosion locations but also as invisible barriers to alongshore sediment transport. Such locations may thus split large

seemingly connected sedimentary systems into smaller cells and compartments. Separation of large sedimentary systems into smaller cells makes it possible to greatly simplify the analysis of properties of the entire system (Kinsela et al., 2017), better understand the functioning and resilience of single compartments, and reach optimum solutions for the design of various structures or beach management and nourishment actions as demonstrated, e.g., in Cappucci et al. (2020) and Susilowati et al. (2022). Moreover, such divergence areas are natural limits for the propagation of pollution that is carried along the shore

with sediment parcels.

Wave-driven sediment transport plays a particularly large role in shaping sedimentary and/or easily erodible shores of relatively young water bodies, such as the Baltic Sea (Fig. 1). Wave impact is almost negligible for the development of its western, northern and north-eastern bedrock coasts that have very little sandy coast. The other shores of this sea, from southern Sweden counter-clockwise to the vicinity of Saint Petersburg is predominantly sedimentary, most of which is still

rapidly developing (Harff et al., 2017). The coastal stretch from the Sambian Peninsula (Samland) to Pärnu Bay in the Gulf of Riga is a >700 km long interconnected sedimentary coastal system, with an almost continuous strip of sand and mostly counter-clockwise sediment transport (Knaps, 1966; see Viška and Soomere, 2013b for references). This transport is particularly massive along the north-western shore of Latvia where it reaches 1,000,000 $m^3yr^{-1}$ (Knaps, 1966).

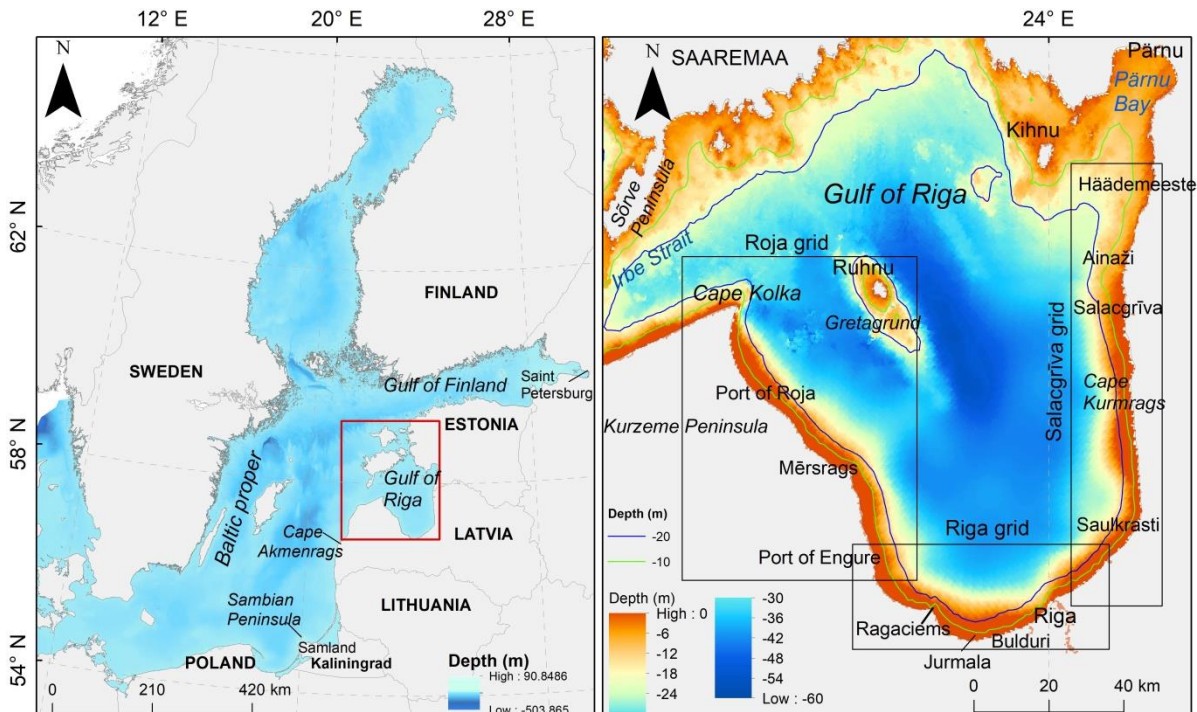

**Figure 1:** Map of the study area (left), showcasing the three subgrids of the wave model used in the analysis of wave-driven alongshore sediment transport (right). The entire left panel represents the area covered by the outermost grid. The area covered by the 2nd level grid (left panel, red box) is chosen to properly represent the wave fields entering the Gulf of Riga directly from the Baltic proper and via the straits connecting this gulf with the West Estonian Archipelago. See detailed bathymetric maps of the Gulf of Riga, e.g., in Tsyrulnikov et al. (2008, 2012).

Wave-driven transport along this stretch of coast was estimated at a relatively low spatial resolution of about 5.5 km 1970–2007 (Viška and Soomere, 2013b; Soomere and Viška, 2014). There is one major accumulation area near Cape Kolka (north-western Latvia) and one almost permanent sediment flux divergence area near Cape Akmenrags on the western coast of Latvia. Both these features are a result of the interplay between the shape and orientation of this stretch of coast and the two predominant wind directions (southwest and north-west or north-northwest) in the area (Soomere, 2003).

These features divide this sedimentary system into three major compartments. Two of them are weakly interconnected with potential annual net sediment flux across Cape Akmenrags occurring approximately once in 40 yr (Soomere and Viška, 2014). Sediment transport from the Baltic proper shores to the interior of the Gulf of Riga is apparently an almost entirely one-way process. The spatial resolution of the transport model used in Soomere and Viška (2014), however, is too low to identify smaller-scale features of alongshore sediment transport and partially or totally separated sedimentary cells. Some indication of their presence can be inferred from the existence of temporary divergence areas and reversals (clockwise transport) of alongshore sediment flux (Viška and Soomere, 2013b) at many locations. These simulations have ignored the presence of man-made structures that may partially or totally block wave-driven sediment transport and thus create additional fragmentation of sedimentary systems.

The focus of our study is the Gulf of Riga where observations of R. Knaps (1966) signal a complicated pattern of erosion, transit and accumulation areas (Fig. 2). Erosion was observed near Roja, south of Mersrags and between Cape Kurmrags and Salacgriva while accumulation was noted to the north of Mersrags, near Jurmala (Bulduri, Fig. 1) and Riga, and near Ainaži. These observations, apparently stemming from the 1960s and updated in the 1990s (Ulsts, 1998), suggest that the actual pattern of sediment transport along the shores of this water body may be quite complicated.

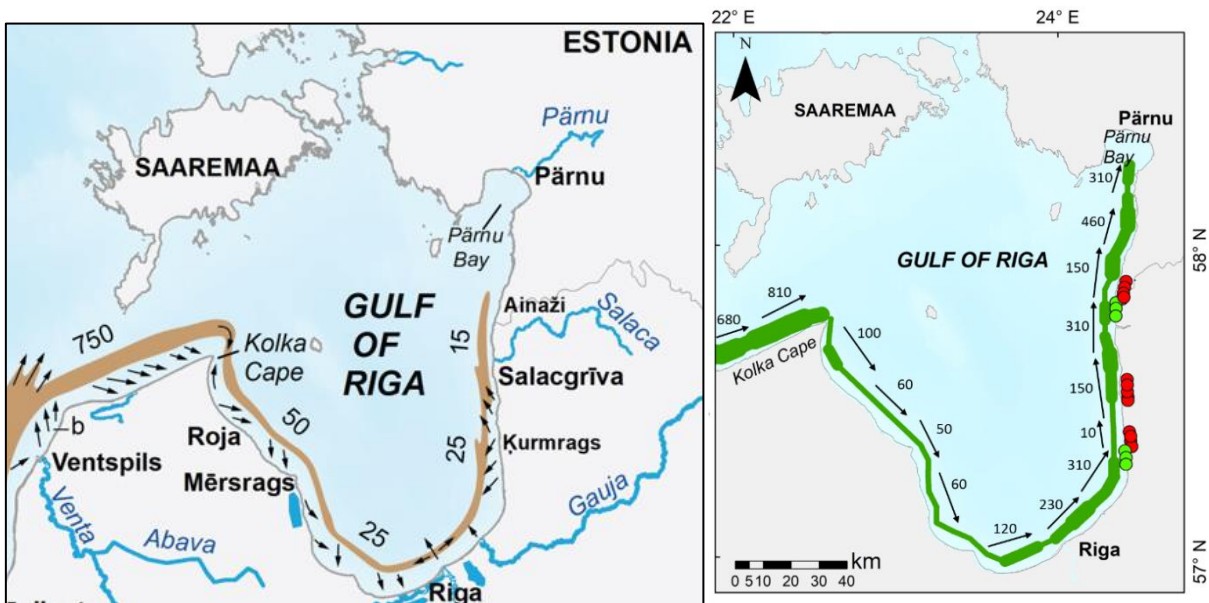

**Figure 2:** Left: Sediment transport into the Gulf of Riga and along its western, southern and eastern shores in thousands m$^3$yr$^{-1}$ evaluated from in situ observations (Knaps, 1966; Ulsts, 1998). Erosion and accumulation areas are indicated with small arrows. Adjusted from graphics created by K. Ehlvest. Right: Numerically estimated wave-driven potential net sediment transport in thousands m$^3$yr$^{-1}$ 1970–2005 (Viška and Soomere, 2013b). Numbers indicate the magnitude of transport, arrows – its direction, red and green circles – sediment flux divergence and convergence areas in single years, respectively.

This conjecture is supported by the analysis of Viška and Soomere (2013b) and Soomere and Viška (2014). They were not able to reproduce minor headlands and smaller changes in the orientation of the coast because of low spatial resolution and thus may have overlooked many local features of transport. Their study suggests that sediment transport along the southern and eastern shores of the Gulf of Riga (Fig. 2) could be much more substantial than estimated in Knaps (1966) and Ulsts (1998). Some of the difference may stem from the limited availability of fine sediment in this part of the study area. More importantly, they highlighted several frequently occurring divergence points of sediment flux and spatially varying temporary reversals of the overall counter-clockwise sediment transport in terms of annual potential net transport between Cape Kurmrags and Saulkrasti (Figs. 1, 2).

These observations match the conclusions of earlier studies (Knaps, 1966; Ulsts, 1998) suggesting that sediment transport may have a discontinuity (a persistent location of divergence of net sediment flux) in the vicinity of Cape Kurmrags. This kind of discontinuity would be impossible in a wave climate where winds from one particular direction (e.g., south-west,

SW) dominate. The transport pattern along any almost straight coastal stretch would then be one-directional. The presence of such a discontinuity is, however, a natural feature of shores that evolve under a two-peak directional distribution of predominant winds (Eelsalu et al., 2023; 2024b). This is the case in the study area where SW and north-western or north-north-western (denoted as (N)NW below) winds predominate (see Section 2.2 for details). Waves generated by (N)NW winds predominate in sediment transport to the south of a certain location in the eastern shore of the Gulf of Riga and waves driven by SW winds predominate to the north of it. A natural conjecture deriving from the large alongshore variation in this location (the divergence area of sediment flux) over >20 km (four grid points of Viška and Soomere, 2013b) between Cape Kurmrags (Fig. 2) and Saulkrasti (Fig. 1) is that several partially isolated sedimentary compartments may exist in this area.

The main objective of this study is to decompose the sedimentary system along a semi-isolated coast in the interior of the Gulf of Riga (Fig. 1), into partially or totally separated sedimentary cells and compartments based on simulations of wave-driven alongshore transport at a considerably increased spatial resolution that matches the typical spatial scale of coastal formations in this region and allows for the identification of man-made features blocking sediment transport. The improved resolution makes it possible to correlate more exactly the directional structure of incoming wave forcing with the bathymetry and geometry in the study area and sheds more light on the associated structure of alongshore sediment transport. Along with a straightforward update of the earlier estimates of potential sediment transport, we aim to more exactly specify sediment flux divergence and convergence areas and the associated configurations of sedimentary compartments and cells on the sedimentary shores of the Gulf of Riga. This analysis is followed by quantification of trends and interannual variations in the sediment transport in this area. Finally, we question why an interesting signal of wave climate change, namely, a permanent increase in bulk transport 1970–2007 from Cape Taran to Pärnu Bay, combined with an increase in net transport 1970–1990 and decrease 1990–2007 (Soomere et al., 2015), was not detected in the Gulf of Riga (Viška and Soomere, 2013a).

As the seabed of the northern and north-eastern parts of this water body from the Sõrve Peninsula to Pärnu Bay (Fig. 1) is rocky or muddy, mostly with low availability of mobile sediment, the shoreline is heavily indented and the shallow area contains numerous islets and underwater features (Tsyrulnikov et al., 2008), we focus on the eastern, southern and western shores of the gulf that comprise an almost continuous sandy strip. These coastal stretches are represented by the Salacgriva grid, the Riga grid and the Roja grid, respectively, in Fig. 1. The northern and north-eastern parts of the gulf are however naturally included in the wave model that covers the entire Gulf of Riga.

We use a set of time series of wave properties derived from a three-level nested SWAN wave model with a spatial resolution of the innermost grids of about 600 m. Wave-driven bulk and net potential sediment transport is evaluated using the Coastal Engineering Research Centre (CERC) approach (USACE, 2002). The results of the analysis are interpreted in terms of annual values of bulk and net potential transport. Section 2 gives an overview of the study area and its wind and wave climate, an insight into how the wave data are obtained and validated, how alongshore sediment flux is evaluated, and how the presence of man-made structures is interpreted. Section 3 presents the analysis of the core properties of sediment flux in different parts of the Gulf of Riga and depicts the division of these shores into sedimentary cells and compartments.

Section 4 highlights similarities and differences of sediment transport on the western, southern and eastern shores and discusses the implications of the established features for coastal processes.

## 2 Method and data

### 2.1 Study area

The Gulf of Riga (Fig. 1) is the third largest semi-enclosed subbasin of the Baltic Sea, with a surface area of 17,913 km$^2$ and an average and maximum depth of 21 and 52 m, respectively. A detailed overview of the basic geographical, geological, climatic and oceanographic features of the Baltic Sea and its larger subbasins are provided in (Feistel et al., 2005; Leppäranta and Myrberg, 2009). It has an oval-like shape with dimensions of approximately 130 × 140 km (Suursaar et al., 2002). As mentioned above, its northern and north-eastern parts have irregular bathymetry and geometry and are not addressed in this study. The bathymetry in the central part of the gulf and in the study area is regular (except for the island of Ruhnu and shallow Gretagrund to the south of this island). The width of the shallow nearshore varies insignificantly. The 10 m and 20 m isobaths are located approximately 2 km and 3.5–8 km from the shore, respectively, along the Latvian shores (see, e.g., Figure 1 in Tsyrilnikov et al., 2012). The main sedimentological properties of the nearshore of the study area are presented in (Viška and Soomere, 2013b).

Similar to the entire Baltic Sea, the coasts of the Gulf of Riga are relatively young and develop comparatively rapidly (Harff et al., 2017; Eelsalu et al., 2025b). They have shown only slow coastal retreat or advance 1935–1990 (Ulsts and Bulgakova, 1998; Eberhards and Lapinskis, 2008). The extent of the eroding areas and the rate of erosion seem to have increased 1992–2007 (Tõnisson et al., 2013), with the fastest erosion seen near Roja, Engure, Ragaciems, and Jurmala in the western and southern parts of the gulf, and near Saulkrasti and Cape Kurmrags in the eastern part (Eberhards and Lapinskis, 2008). See also Luijendijk et al. (2018) for the latest estimates.

According to Bertina et al. (2015), "[the] Gulf of Riga is an area in which combined sea erosion and accumulation processes, as well as alluvial processes, play significant roles in the coastal development." They reported relatively rapid coastline retreat immediately to the south-east of the western jetty of the Daugava River mouth and equally rapid coastline advancement further south-west until the Lielupe River mouth. Coastal processes are fastest during strong (wave) storms when accompanied by a high water level, such as in hurricane Erwin/Gudrun in 2005 (Eberhards et al., 2006; Lapinskis, 2017), during which the maximum shoreline retreat was 15–27 m. This storm most strongly affected southern and eastern coasts of the Gulf of Riga (Eberhards et al., 2006) and an estimated 0.8 million m$^3$ of sediment was lost from the subaerial part of the coastal slope (Lapinskis, 2017).

The described processes are obviously related to unusually massive wave-driven alongshore sediment transport in the eastern Baltic Sea under relatively mild wave conditions (see Section 2.2 for more details). About 700,000–800,000 m³ of sand is transported per year towards Cape Kolka along the north-western shore of Kurzeme Peninsula (Knaps, 1966; Viška and Soomere, 2013b; Jankowski et al., 2024) (Fig. 2). About 90 % of this mass is deposited in the vicinity of Cape Kolka

and only about 50,000 m³yr⁻¹ is further transported into the sedimentary system of the Gulf of Riga (Knaps, 1966). This transport is almost entirely one-way. The accumulation area is to the north of Cape Kolka as the eastern shore of the cape is rapidly eroding (Fig. 3). The magnitude of sediment transport evaluated from observations is from 15,000 to 50,000 m³yr⁻¹ in different segments of the study area (Knaps, 1966; Ulsts, 1998). It is much smaller near Riga and remained undefined for the coastal segment to the north of Ainaži in older estimates (Knaps, 1966; Ulsts, 1998) while lower-resolution simulations (Viška and Soomere, 2013b) suggested that potential net transport flux continued almost unidirectionally towards Pärnu.

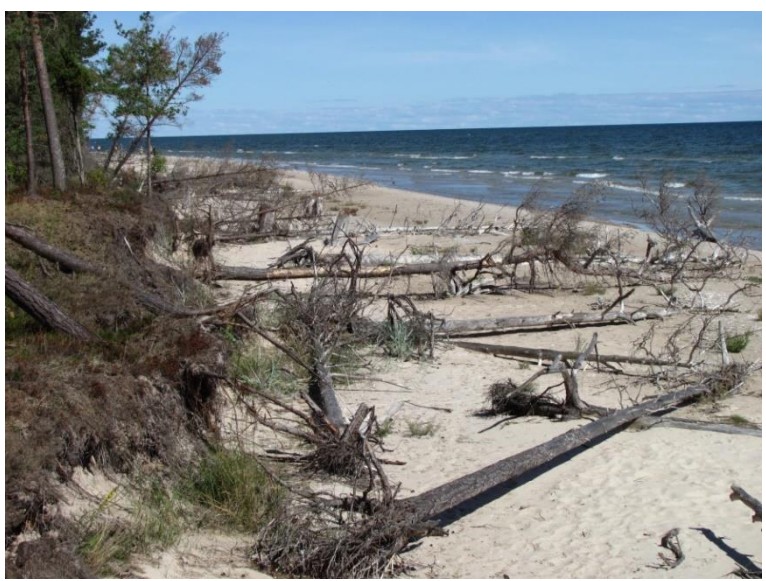

**Figure 3:** Eroding eastern shore of Cape Kolka. Photo by T. Soomere, 24.08.2013.

## 2.2 Wind and wave climate in the study area

The study area is located at the southern margin of the North Atlantic storm track. It is characterised by the frequent passage of low pressure systems from the North Atlantic that often produce high winds that are favourable for both severe wave generation (Björkqvist et al., 2017, 2020) and wind energy generation (Barzehkar et al., 2024). The area of relatively persistent high winds (in terms of the capacity factor, Barzehkar et al., 2024) extends from the SW part of the Baltic Sea to the nearshore of Latvia and Estonia, and also embraces the Gulf of Riga.

This situation gives rise to a highly anisotropic wind climate that is much more complicated than simply a dominant air flow from the west. While the directional distribution of weaker winds is almost isotropic, moderate and strong winds mostly blow from two directions in the north-eastern Baltic proper (Soomere and Keevallik, 2001; Soomere, 2003). The majority of such winds blow from the SW while winds from the (N)NW directions are less frequent but may have even larger speeds than the SW winds (Soomere, 2001). This two-peak pattern of predominant moderate and strong winds is characteristic for the study area. It is less evident at the latitudes of the Gulf of Finland and to the south of Lithuania (Soomere et al., 2024).

This strongly anisotropic pattern, with relatively weak and infrequent easterly winds, is evidently responsible for very high water levels in two locations of the Gulf of Riga: Pärnu in the north-east and Riga in the south-east of the gulf (Hünicke et al., 2015; Männikus and Soomere, 2023).

Similar to the entire Baltic Sea, wave fields in the Gulf of Riga are almost entirely driven by local storms and contain a small proportion of long-period swell (Björkqvist et al., 2021; Najafzadeh et al., 2024). This feature means that long-term average significant wave heights are fairly low, well below 1 m in the Gulf of Riga and even the higher percentiles remain moderate (Fig. 4), but unexpectedly severe wave conditions may occur in this basin (Björkqvist et al., 2017; Najafzadeh et al., 2024). In other words, the wave climate is highly intermittent (Soomere and Eelsalu, 2014) in the sense that most of the annual wave energy arrives the coast in a few days. Consequently, the propagation direction of waves during these storms plays the most important role in coastal evolution.

As the wave fields of substantial height are fetch-limited (that is, their properties and most importantly wave propagation direction largely follow the local wind properties) in the Gulf of Riga (Najafzadeh et al., 2024), waves excited by predominant strong winds from the SW or (N)NW play a key role in coastal processes and alongshore sediment transport in this water body. More specifically, waves from these narrow ranges of direction often provide up to 80–90% of the total net and bulk transport. In other words: what happens in a particular coastal location largely depends on the delicate balance of alongshore transport under the impact of these two wave systems (Eelsalu et al., 2024b).

## 2.3 The SWAN model data for the nearshore of the study area

The instantaneous rate of wave-driven potential alongshore sediment transport is evaluated using the classic Coastal Engineering Research Council (CERC) approach (USACE, 2002). This model relates sediment transport in the nearshore, from the breaker line to the shoreline, with the arriving wave energy flux at the breaker line, water and sediment density, and sediment porosity under the assumption of unlimited availability of non-cohesive sediment. For this purpose we employ a high-resolution time series of significant wave height, average wave propagation direction and peak period reconstructed for the time period 1990–2022 using a triple nested version of the third-generation phase-averaged spectral wave model SWAN (Booij et al., 1999). The model cycle III, version 41.31A was forced by ERA5 wind information (Hersbach et al., 2020) in an idealised ice-free set-up. The presence of currents and varying water levels was ignored. Varying water levels do not affect our results because we only consider the idealised case of potential transport that is independent of the particular water level. The main limiter of the accuracy of calculations is the quality of wind and bathymetry information. The presence of currents may modify wave properties to some extent but there is currently no way to reliably replicate the current system of the Gulf of Riga. Ignoring ice cover apparently leads to an overestimation of transport of up to 20% (Najafzadeh and Soomere, 2024).

A detailed overview of the particular wave model implementation and its validation for the Baltic proper and Gulf of Finland against instrumentally recorded wave data is provided in Giudici et al. (2023). The quality of the reconstruction of wave properties in simulations using ERA5 winds in this basin of fairly complicated shape was generally better than in simulations using local high-quality open sea winds (Männikus et al., 2024). An additional verification of the output of the

model in the Gulf of Riga and near its entrance in the eastern Baltic proper as well as a thorough description of the Gulf of Riga wave climate 1990–2021 is provided in Najafzadeh et al. (2024) and briefly summarised in Section 2.2.

This model is applied to the entire Baltic Sea at a 3 nautical mile (nmi) resolution and to the Gulf of Riga and its vicinity (Fig. 1) at a 1 nmi resolution (0.03° in the East-West direction and 0.015° in the North-South direction). The eastern, southern and western coastal areas of the gulf, with a mostly straight shoreline, are covered with three realisations of a regular rectangular grid with a resolution of 0.32 nmi (about 600 m) called the Roja, Riga and Salacgriva grids, respectively (Fig. 1). The extent of the sets of relevant grid points along the shoreline, from which input information for transport calculations is retrieved, is indicated in Fig. 4. Accordingly, the shoreline of the study area is divided into about 600–800 m long sectors depending on the mutual orientation of the shoreline and grid cells. The grid system employs a one-way information flow of wave properties from the 3 nmi grid to the 1 nmi grid and then separately to each of the three 0.32 nmi grids (Najafzadeh et al., 2024). Simulations of wave properties on the innermost 0.32 nmi grids are performed independently.

An adequate application of the CERC approach presumes that wave properties are known somewhere offshore from the breaker line (USACE, 2002). This is a challenge for high-resolution wave models that extend almost to the shoreline. Several grid points of the wave model close to the shore have a water depth of only 1–2 m. Small waves that are adequately described by the model at such depths may serve as an important constituent of the sediment transport system in this area (Eelsalu et al., 2022). However, most sediment motion is usually generated by a few of the strongest storms in the year (Różyński, 2023). As mentioned above, the wave climate of the eastern Baltic Sea is extremely intermittent: some 30 % of the annual wave energy flux arrives within a few days with very severe waves (Soomere and Eelsalu, 2014). Wave properties for the evaluation of wave-driven transport using the CERC model should be taken from those model grid cells that adequately reflect the most severe wave conditions. Such grid cells are normally located offshore of the breaker line that exists in the strongest storms.

A natural limit for water depth at which the breaker line is located is the closure depth, down to which strong waves systematically relocate sediment. The closure depth, evaluated using wave data with 5.5 km resolution for 1970–2007 (Räämet and Soomere, 2010; Soomere and Räämet, 2011, 2014), varies between 3 and 5 m (Soomere et al., 2017), being the largest near Pärnu and in the Irbe Strait. This resolution obviously does not resolve many important features of nearshore bathymetry and shore geometry in the study area.

To more adequately represent the properties of severe wave storms for the CERC model, we selected wave model grid cells for calculations of wave-driven sediment transport based on the 95th percentile wave heights, bathymetry data and re-estimated closure depths (Fig. 4). More specifically, we employed a four-step procedure for this selection. Firstly, we identified the closest cells along the shoreline that had water depth at least twice the 95th percentile of simulated significant wave height for each coastal segment. Secondly, we re-evaluated closure depth for these cells (Fig. 4). Thirdly, wave simulations with a resolution of ~600 m for 1990–2022 used in this paper add several nuances to this pattern. They stressed the severity of waves in the south of the gulf near Riga and also showed that the values of closure depth at this resolution do

not necessarily match similar values estimated using a lower resolution (Soomere et al., 2017). As the SWAN model adequately resolves the loss and redistribution of wave energy in relatively shallow water, the closure depth estimated at this resolution may considerably depend on the water depth in a particular grid cell. For this reason, closure depth was re-evaluated for the selected cells. Based on this estimate (Fig. 4), the initially selected cell was replaced by the adjacent cell closer to or further from the coast, having in mind that the water depth in the finally selected cell should generally exceed the closure depth evaluated for each particular location. As the fourth step, this selection was on some occasions adjusted to mirror the overall coastline shape with the set of selected grid cells and, where applicable, to maintain a more or less constant distance from the coastline.

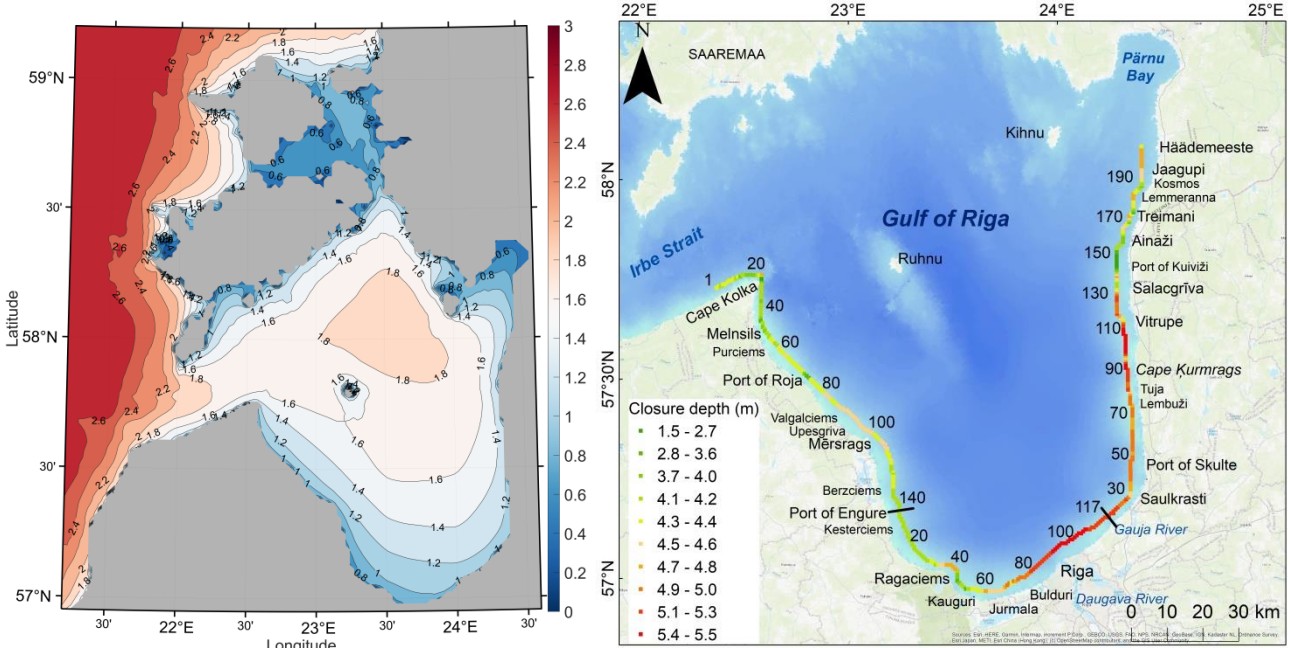

**Figure 4**: Left: 95[th] percentile of significant wave height in the Gulf of Riga and its vicinity based on the SWAN model simulations 1990–2022 with a resolution of 1 nmi (Fig. 1). Right: Closure depth (colour code) at wave model grid points and sequential numbers of grid cells selected for the study from three model grids (Roja, Riga and Salacgriva grids, Fig. 1) based on wave data from Giudici et al. (2023) and Najafzadeh et al. (2024). Short black lines on the right panel: separation of the model grids with 600 m resolution.

The set of selected wave model grid cells (Fig. 4) contains 159 cells along the western shore of the Gulf of Riga (Roja grid, Fig. 1), 117 cells along the southern shore (Riga grid) and 201 cells along the eastern shore (Salacgriva grid). Each such cell was associated with the average orientation of the coastline and isobaths down to the closure depth. In essence, the coastline of the study area was approximated with a piecewise straight line consisting of lines with this orientation (Fig. 5). The length of such pieces usually varies between 560–800 m depending on the orientation of the coastline with respect to coordinate lines. Some cells were in the overlapping parts of the grids. The natural boundaries of grids were at the Port of Engure and at the Gauja River mouth (Fig. 4). These locations are major obstacles for wave-driven alongshore sediment transport. The analysis below includes 22 cells and associated coastal sectors to the west of Cape Kolka (to provide an

280 indication of transport along the Baltic proper shore) and 123 cells from Cape Kolka to the Port of Engure in the Roja grid, 110 cells from the Port of Engure to the Gauja River mouth along the southern shore of the Gulf of Riga, and 190 cells from the Gauja River mouth to the Estonian township of Häädemeeste along the eastern shore of the gulf.

## 2.4 Wave properties at the breaker line

In situations where waves usually approach the shore at a small angle between the wave propagation direction and shore
normal it is reasonable to evaluate changes in wave properties from the selected wave model grid cells to the breaker line by means of evaluation of wave shoaling and loss of wave energy due to whitecapping and wave-bottom interaction using simplified formula (e.g., Larson et al., 2010). The situation is more complicated in the Baltic Sea where waves often approach the shore at large angles (Eelsalu et al., 2024a). Waves in the Gulf of Riga are usually shorter than in the Baltic proper (Eelsalu et al., 2014; Najafzadeh et al., 2024). This feature together with the narrowness of the relatively shallow
nearshore (see above, Fig. 1) implies that the impact of refraction on wave propagation is usually comparatively small and wave fields frequently approach the breaker line in the Gulf of Riga at a relatively large angle.

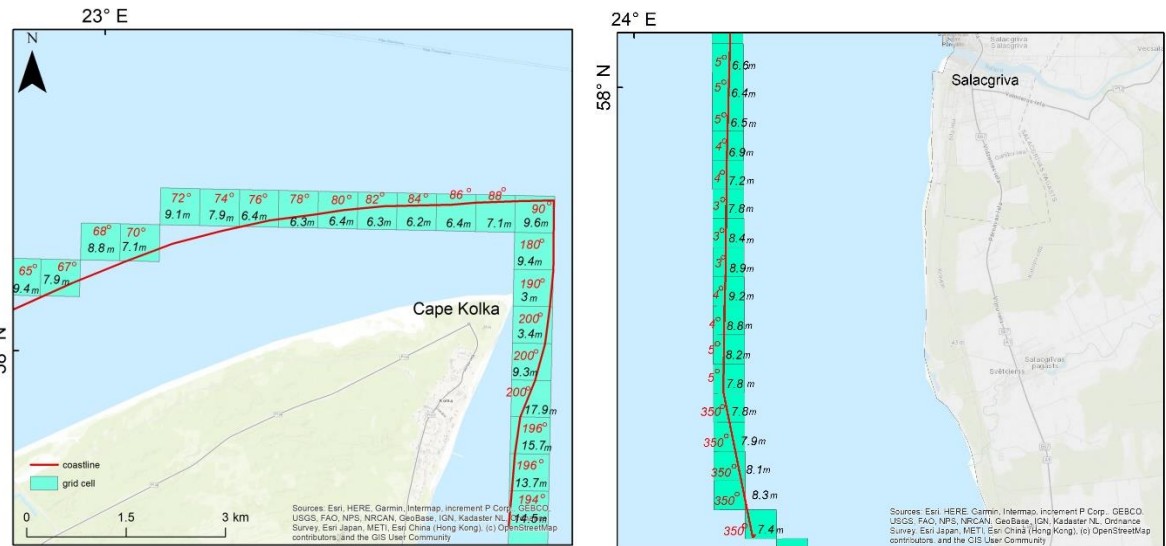

**Figure 5:** Examples of wave model grid cells used in the analysis, water depth in these cells and the associated orientation of the coastline (bold line in the cells) near Cape Kolka (left) and Salacgriva (right).

In this case it is necessary to evaluate the joint impact of shoaling and refraction to wave properties along the path of the waves from the model grid cell to the breaker line. This can be done, to a first approximation, by assuming that the seabed height increases smoothly shoreward from the wave model grid cell to the breaker line, with isobaths parallel to the shoreline. This assumption, even though not perfect, makes it possible to analytically evaluate the joint effect of shoaling and refraction on the properties of the waves that approach the shore at a relatively large angle, during their propagation from the
nearshore model grid cells to the breaker line. In the idealised case of a monochromatic wave field with a height $H_0$ that

propagates towards the shore with a phase and group speed $c_{f0}$ and $c_{g0}$, respectively, and at an angle $\theta_0$ between the wave vector and shore normal, an application of linear wave theory leads to the following algebraic equation of 6th degree for the wave height $H_b$ at the breaker line (Soomere et al., 2013; Soomere and Viška, 2014):

$$H_b^5 g \left(1 - \frac{H_b g}{\gamma_b} \frac{\sin^2 \theta_0}{c_{f0}^2}\right) = H_0^4 \gamma_b c_{g0}^2 (1 - \sin^2 \theta_0). \tag{1}$$

The subscript "$b$" denotes the wave properties at the breaker line. A simple way to close Eq. (1) is to assume that (a) the breaking index $\gamma_b = H_b/d_b = 0.8$ is constant (where $d_b$ is the water depth at the breaker line) and (b) breaking waves are long waves, which means that $c_{gb} = c_{fb} = \sqrt{gd_b}$ at the breaker line. These approximations are not perfect: the breaking index may substantially vary (Lentz and Raubenheimer, 1999; Power et al., 2010; Raubenheimer et al., 1996, 2001; Sallenger and Holman, 1985) and breaking waves are often not ideal long waves. An advantage of these assumptions is that

they make it possible, to first approximation, to systematically take into account specific features of wave fields that approach the shore at a large angle. The smaller of the two real solutions of Eq. (1) indicates the breaking wave height. The angle between the wave propagation direction and shore normal at breaking is evaluated using Snell's law.

**2.5 Evaluation and interpretation of sediment transport**

The hourly values of instantaneous potential sediment transport is evaluated for each coastal sector associated with the

relevant selected wave model grid cell using the CERC approach (USACE, 2002) based on hourly time series of wave properties at the breaker line. The core approximation in the CERC formula $I_t = KP_t = KEc_{gb} \sin \theta_b \cos \theta_b$ is that the wave-driven transport rate is proportional to the rate of beaching of the wave energy flux $Ec_g$ ($E$ is the wave energy at the breaker line) in the given coastal sector. The quantity $I_t = (\rho_s - \rho)(1 - p)Q_t$ has the meaning of the potential immersed weight transport rate that is proportional to the potential alongshore sediment transport rate $Q_t$ (USACE, 2002), $\rho_s$ is the

density of non-cohesive sand, $p$ is the porosity coefficient, and $\rho$ is water density. The transport was interpreted as positive (counter-clockwise drift) if it was directed to the right with respect to the observer looking to the sea. The net transport for a coastal sector and a specific time period was evaluated as the sum of directional values of hourly transport, that is, taking into account the sign of $Q_t$. This quantity mirrors the amount of sand that would be actually transported along the shore during a certain time interval in ideal conditions. The bulk transport was calculated as the sum of absolute values of $Q_t$,

equivalently, as an integral of the absolute value of instantaneous transport over the period of interest, from single months to the entire simulation period. This quantity provides an estimate of the total amount of sand that was moved in the sector in any direction, including 'back-and-forth' transport in ideal conditions.

     We use constant values of porosity coefficient $p = 0.4$ and water density $\rho = 1004$ kg m$^{-3}$ that roughly correspond to the typical material of sand (quartz) and the average salinity of 4.90–5.38 g kg$^{-1}$ of the upper mixed layer of the Gulf of Riga

(Skudra and Lips, 2017). We employ the direction-depending expression $K = 0.05 + 2.6 \sin^2 2\theta_b + 0.007 u_{mb}/w_f$ for the CERC coefficient $K$ (USACE, 2002). Here $u_{mb} = (H_b/2)\sqrt{g/d_b}$ is the maximum orbital velocity in linear waves, $w_f =$

$1.6\sqrt{gd_{50}(\rho_s - \rho)/\rho}$ is the fall velocity. We assume that the typical grain size $d_{50} = 0.17$ mm is constant and apply the density of sand $\rho_s = 2650$ kg m$^{-3}$.

While the modelled wave time series were carefully validated against several sets of recorded wave properties (Giudici et al., 2023; Najafzadeh et al., 2024), similar validation of evaluated transport rates against direct observations of transport was not possible because of the absence of contemporary field data. For this reason, the validation was performed implicitly, by means of comparison of the results with earlier observations (Knaps, 1966; Ulsts, 1998), the output of lower-resolution simulations (Soomere and Viška, 2014), and otherwise known areas of erosion or accretion. However, as the simulated potential transport reflects the wave impact on coastal sediment in ideal conditions of unlimited availability, actual transport is usually much less intense.

The most interesting coastal segments are the locations of the zero-crossings of net transport. The upcrossings in this projection (positive transport direction to the right with respect to the observer looking to the sea) indicate divergence points of sediment flux and thus serve as most likely erosion areas (Fig. 6) and natural barriers separating sediment cells. The downcrossings are convergence points of sediment flux that usually mirror accumulation areas. In a similar manner, an increase in alongshore net transport from the left to the right usually reflects locations with sediment deficit and a decrease in this transport in this direction reflects accumulation regions (Fig. 6).

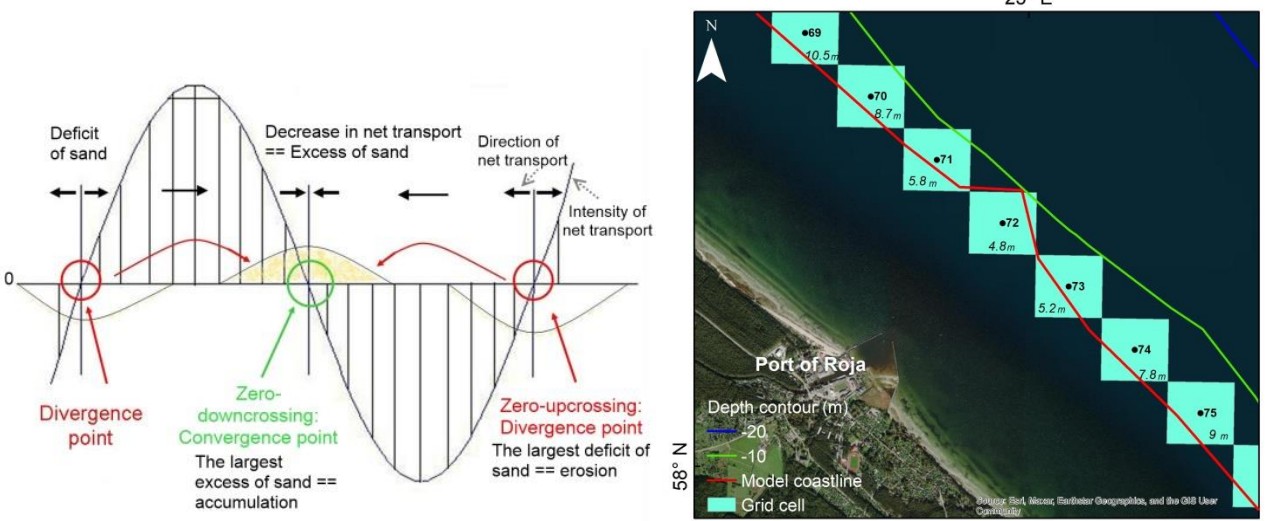

**Figure 6:** Left: A schematic for the interpretation of alongshore changes of the intensity and direction of wave-driven alongshore net transport areas (Soomere and Viška, 2014). Reprinted with permission from Elsevier, Licence 5850280172485. Right: location of wave model grid cells and orientation of the model coastline in the vicinity of the Port of Roja.

The resolution of wave and sediment transport models is such that the presence of breakwaters at major river mouths and several smaller harbours is reflected in the location of the grid cells that are used to evaluate wave properties at the breaker line as well as in the local orientation of the breaker line (Fig. 6). Breakwaters of such harbours usually extend to 300–500 m or even more offshore from the coastline into water depths that exceed the closure depth. This is the situation at Roja,

Mersrags, Engure, the Daugava River mouth, Skulte, Salacgriva, Kuiviži, Ainaži, Treimani and the Kosmos establishment in Estonia. On the one hand, such structures almost totally block wave-driven alongshore sediment transport, most of which occurs in the surf and swash zone. On the other hand, sediment accumulation at the downstream (or outer) side of such structures leads to a rapid variation of the orientation of shoreline and isobaths near the structure. The piecewise linear approximation of the shoreline and isobaths described in Section 2.4 largely follows the orientation of the breakwaters or jetties in the relevant cells and thus has substantially different orientation than its neighbouring sections (Fig. 6). The formal application of the CERC approach usually leads to completely unrealistic estimates of sediment transport in such cells. It is therefore natural to remove such locations from calculations of alongshore transport. Moreover, it is also natural to conclude that structures that extend deeper than closure depth serve as almost complete barriers to sediment flux in the sense that waves and associated currents may transport some sediment around them in extreme conditions but no through transport occurs under usual conditions.

A direct consequence of the use of CERC approach is that we only evaluate alongshore transport. This approximation is partially justified in the light of the presence of unusually strong alongshore transport in the study area under the relatively mild wave climate. The main reason for such intense transport is that waves often approach the shore at a large angle. A natural consequence of this feature is that cross-shore transport usually plays much smaller role than might be expected in most of the eastern Baltic Sea shores, except for a few locations (e.g., Šakurova et al., 2025). An implication of neglecting cross-shore transport is that shoreline relocation does not necessarily follow the accumulation or erosion rates. However, our conclusions only concern alongshore variations in the wave-driven transport and the impact of man-made structures to this transport, and thus are invariant with respect to the impact of cross-shore transport.

## 3 Alongshore sediment transport patterns

### 3.1 Almost unidirectional transport along the western shore

We start the analysis from the western shore of the Gulf of Riga that is represented by the Roja grid in Fig. 1 and is defined to extend from the area of Cape Kolka to the Port of Engure (Fig. 4). An extension of the study area to the north-western shore of Cape Kolka over about 15 km (22 wave model grid points, Fig. 5) provides an option to compare the results with in situ observations and earlier simulations. As expected, the intensity of potential wave-driven bulk (independent of direction) sediment transport in the interior of the Gulf of Riga is several times smaller than along the Baltic proper shore of Latvia (Fig. 7). While the typical bulk transport is about 1,000,000 $m^3yr^{-1}$ to the west of Cape Kolka (Viška and Soomere, 2013b; Jankowski et al., 2024), it drops to 200,000±100,000 $m^3yr^{-1}$ to the east of the cape, with only one short segment of transport of 300,000±200,000 $m^3yr^{-1}$ around a headland near Mersrags. These quantities are also typical of the southern shore of the gulf as will be discussed below.

The sediment transport direction is predominantly counter-clockwise (positive in our framework, Fig. 7, middle panel), that is, to the south-east along the western shore of the gulf. Different from many locations on the Baltic proper shores

(Viška and Soomere, 2013b; Eelsalu et al., 2024a) or in the vicinity of Tallinn Bay on the northern shore of Estonia (Eelsalu et al., 2023), transport in the opposite (clockwise) direction (a reversal) has a considerable role between Purciems (Fig. 4) and the Port of Roja, and also between Upesgriva and Mersrags, and in some years on the eastern shore of Cape Kolka 390  (Fig. 7, lower panel). The latter feature is consistent with historic in situ observations (Knaps, 1966; Ulsts, 1998). The former features are not indicated in historic observations. All three reversals evidently have been smoothed out in earlier lower-resolution simulations (Viška and Soomere, 2013b).

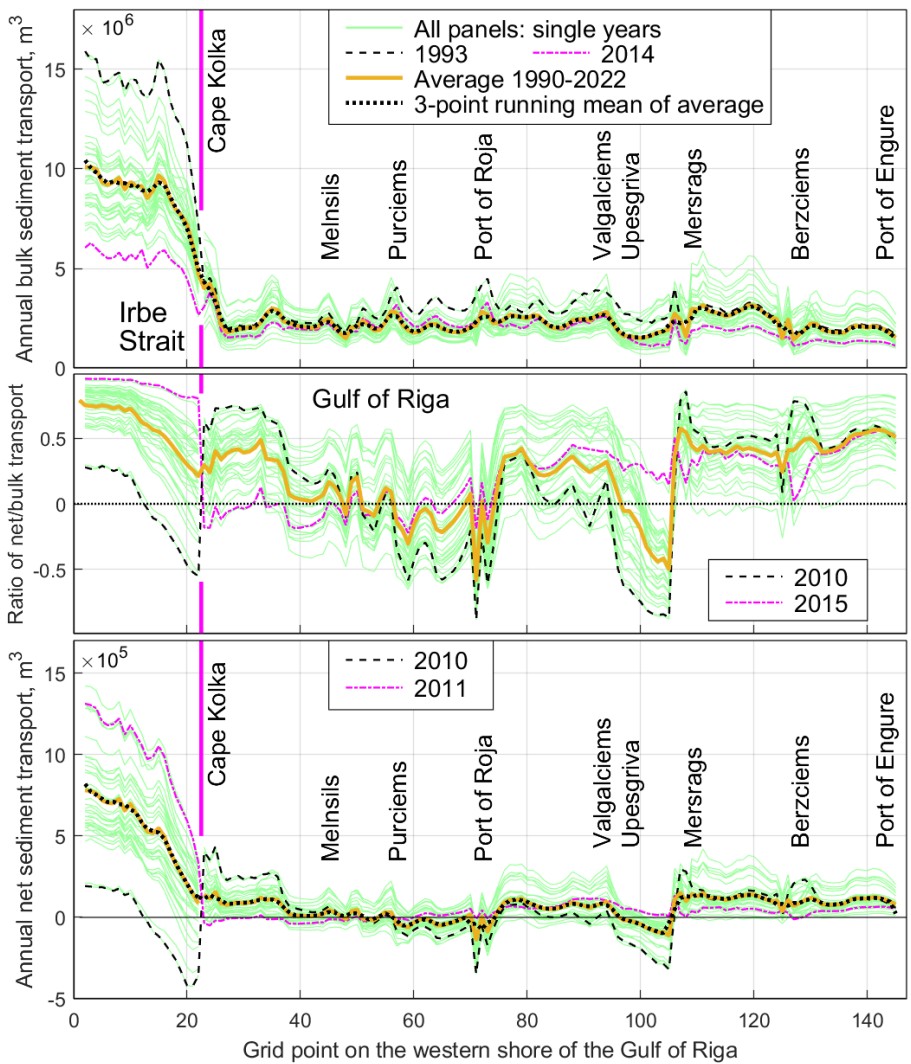

**Figure 7:** Simulated wave-driven potential bulk sediment transport (upper panel), ratio of net to bulk transport (middle panel) and net 395  potential sediment transport (lower panel) along the western shore of the Gulf of Riga. The blue line (average transport 1990–2022) is almost wholly masked by the red line (3-grid point running mean of the average transport 1990–2022) in the upper and lower panels. The data for grid points that follow the orientation of breakwaters of the Port of Engure are omitted. See locations in Fig. 4 and a map of the transport scheme in Fig. 13 below.

While the shoreline between the eastern side of Cape Kolka and Roja is locally almost straight and gently curving, the water depth in the nearshore of this shoreline contains extensive alongshore variations in selected wave model grid cells. The most significant feature is an up to 40 m deep area a few kilometres to the east of Cape Kolka (Fig 1; see also a higher-resolution map in Tsyrulnikov et al., 2008). This deep area becomes evident as a water depth of 14–18 m in several wave model grid cells located less than 1 km from the shoreline (Fig. 5, left panel). The 5 m and 10 m isobaths meander noticeably between Cape Kolka and Roja (see, for example, https://fishing-app.gpsnauticalcharts.com/i-boating-fishing-web-app/fishing-marine-charts-navigation.html). This bottom structure apparently reflects streamlined topographical features in the area stemming from Late Weichselian glacial dynamics (Tsyrulnikov et al., 2008) and possibly a different orientation of ice-shaped features at a large angle with respect to the contemporary shoreline in this region during certain stages of the presence of the Fennoscandian ice-sheet (Karpin et al., 2023). This leads to considerable variations in the water depth in grid cells selected for the analysis at a scale of 1–2 km. These dissimilarities translate into local differences in the transport rates and the ratio of net and bulk transport (Fig. 7) because of reasons explained in Section 2.3. However, the properties of net transport are less affected and the "impact" of the described feature is almost lost when averaging over three adjacent grid points.

A discontinuity in the ratio of net and bulk transport to the west of Mersrags mirrors the presence of a headland with abruptly changing orientation of the shoreline. Still, it is likely that, at least in some years, the overall counter-clockwise sediment transport carries sand around this headland to the south-east as the values of net transport are positive along the entire shore of this headland (Fig. 7).

The pattern of the magnitude of annual net sediment transport reinforces and provides detail to these conjectures. The typical rate of counter-clockwise net transport on the Baltic proper shore of Cape Kolka varies from 300,000 to 900,000 $m^3yr^{-1}$, depending on the particular coastal section, with an average of about 600,000 $m^3yr^{-1}$ over a 10 km long stretch to the west of the cape (Fig. 7). This projection matches the outcome of earlier in situ observations (Knaps, 1966; Ulsts, 1998) and simulations (Soomere and Viška, 2014).

The properties of bulk and net transport vary significantly in different years. The years characterised by very intense (e.g., 1993) or very low (e.g., 2014) bulk transport along the north-western shore of Cape Kolka are not mirrored along the coastal stretch to the east of Cape Kolka. The correlation coefficient of bulk transport in single years over all 22 grid points to the west of this cape and 22 points to the east of this cape is –0.14 ($p = 0.43$). The same feature is evident for the ratio of net and bulk transport (years 2010 and 2015 in the middle panel of Fig. 7) and for the net transport. The characteristic feature of the net transport is that years with strong counter-clockwise transport to the west of Cape Kolka (e.g., 2011) correspond to almost zero counter-clockwise transport in the western Gulf of Riga. The similar correlation coefficient for net transport in single years, is –0.68, with $p < 0.0001$, indicating statistically significant negative correlation between these values.

Interestingly, if the net transport and the net/bulk transport ratio have a maximum in some years west of the cape, these quantities have a minimum to the east, and vice versa (Fig. 7). The change in the sign of the net transport at Cape Kolka in years with strong clockwise transport along the western shore of the cape (e.g., 2010) evidently reflects the changing role of

the predominant northerly (N)NW and SW winds in such years. For example, (N)NW winds move sand to the south along both shores of the cape. This transport is negative (clockwise) on its western shore and positive (counter-clockwise) on its eastern shore. Therefore, a major jump and sign change in some annual values of the net transport and the ratio of net and bulk transport at Cape Kolka (highlighted for year 2011 in Fig. 7) naturally reflects years with predominant northerly (N)NW winds. In contrast, counterclockwise transport (positive to the north-east) along the western shore of Cape Kolka is driven by westerly winds. These winds create similar transport to the north (clockwise, negative) along the eastern shore of this cape. As waves created by SW winds have short fetch for the eastern shore of Cape Kolka, clockwise transport created by such waves is fairly weak as exemplified by year 2010 in Fig. 7. Interestingly, there is no jump or discontinuity in the average bulk transport at this location. Another interesting feature is that the ratio of the net and bulk transport may considerably vary with respect to the average value of this ratio in single years (e.g., 2010 and 2015 in Fig. 7).

The intensity of potential net transport varies considerably along the western shore of the Gulf of Riga. Its average magnitude from Cape Kolka to Engure is about 50,000 $m^3yr^{-1}$, and this is consistent with previous findings (Knaps, 1966; Ulsts, 1998; Viška and Soomere, 2013b; Soomere and Viška, 2014). The presence of a zero-downcrossing of net transport in some years immediately to the east of Cape Kolka (around cell #25) mirrors the presence of an erosion area in this location (Fig. 3). Even though there are several locations of relatively frequently occurring pairs of zero-downcrossings between Cape Kolka and the headland near Mersrags, this coastal segment most likely forms a continuous sedimentary system in which sand can move along the entire segment in different years. The shoreline of this area is slightly curved and several sand bars exist in the nearshore along the entire section. Small-scale fluctuations in the numerically evaluated bulk sediment transport and reversals of net transport between Cape Kolka and Roja apparently stem from the choice of particular locations of selected wave model grid cells.

Sharp variations in the ratio of net and bulk transport near Roja reflect the presence of the port and breakwaters. They extend to about 5 m water depth (https://www.gpsnauticalcharts.com/main/latvia/lv613340-port-of-roja-nautical-chart.html, last accessed 26.12.2024) whereas closure depth is below 4 m in this location. It is thus likely that these breakwaters and the >6 m deep entrance channel largely stop alongshore sediment flux. Technically, this feature is reflected by a local reversal of net sediment transport and unrealistic values of net transport and the ratio of net and bulk transport in coastal sectors corresponding to wave model grid cells #71 and 72 (Fig. 7) where the presence of breakwaters affects the orientation of the shoreline approximation in these grid cells (Fig. 6). This rapidly changing orientation actually means that the resolution of the model is not sufficient to replicate sediment transport properties near such structures (see Section 2.5). Similar effects occur in the vicinity of other harbours in the study area and usually also in the estimates of bulk transport. For this reason the estimates of transport in the vicinity of such structures are ignored in the analysis below and are mostly not represented in Figs. 7, 9, and 10.

Relatively intense net sediment transport evidently takes place between Roja and a headland to the west of Mersrags. The impact of a few small-scale headlands and jetties at grid cells #74 and #79 interrupts the continuous sand beach and partially stops sediment transport. Their presence is not reflected in the model. A major headland to the north of Mersrags almost

completely stops the counter-clockwise transport. The orientation of the coastline changes by about 80 degrees. This feature is visible in Fig. 7 as a reversal of net sediment transport in most years. It is therefore safe to say that even in the absence of harbours and breakwaters the coastal segment from Cape Kolka to the headland at Mersrags formed an almost isolated sedimentary compartment in the past that was to some extent fed by sand from the vicinity of Cape Kolka.

A direct consequence is that there is almost no sand on the eastern side of this headland and also in the vicinity of the Port of Mersrags. The water depth of the entrance channel of the Port of Mersrags is >5 m ([https://fishing-app.gpsnauticalcharts.com/i-boating-fishing-web-app/fishing-marine-charts-navigation.html?title=Port+of+Mersrags+boating+app#15/57.3345/23.1406](https://fishing-app.gpsnauticalcharts.com/i-boating-fishing-web-app/fishing-marine-charts-navigation.html?title=Port+of+Mersrags+boating+app#15/57.3345/23.1406)) and the north mole of this port extends to about 4 m deep area. It is thus likely that this port almost fully stops sediment transport for the same reasons as discussed for the Port of Roja even though this feature is not resolved in our simulations. The sandy beach becomes evident again about 10 km to the south of Mersrags as visible, e.g., from Google Earth.

The coastal stretch between Mersrags and Engure also contains a few minor headlands that to some extent modulate the intensity of both bulk and net transport, and their ratio. Different from the above, this stretch has almost entirely (in terms of annual means) counter-clockwise sediment transport. Reversals occur only in a couple of years. The water depth of the entrance channel to the Port of Engure is >4 m (https://www.eastbaltic.eu/engure-marina/). Breakwaters of this port extend even further from the shoreline than those of the Port of Roja and Port of Mersrags into clearly deeper water than closure depth (>3.5 m in this location). It is therefore likely that breakwaters of the Port of Engure (not shown in Fig. 7) and accretional features at these breakwaters almost fully stop the wave-driven sediment transport. Together with the headland at Mersrags they separate this coastal stretch into an almost isolated sedimentary compartment.

## 3.2 Variable transport and accumulation along the southern shore

The southern coast of the Gulf of Riga, represented by the Riga grid in Fig. 1 and defined to extend from the Port of Engure to the Gauja River (Fig. 4), changes its orientation from the north-south direction at Engure (Fig. 4) to the west-east direction near Jūrmala and to the south-west-north-east alignment near the Gauja River mouth (Fig. 1). This pattern of changes means that the largest driver of sediment transport between Engure and Jūrmala are waves generated by (N)NW winds while the predominant driver near Riga (Daugava River mouth) and further to the east are SW winds, the fetch length of which increases from the west to the east. The coastline is smoothly curved from Engure to Ragaciems (Fig. 4), with a gentle headland at Ragaciems, and is again gently curved from Ragaciems to the Daugava River mouth and to the north-east of the Daugava River mouth. The massive breakwaters at the river mouth (Fig. 8) almost completely stop wave-driven alongshore transport and divide the coastal stretch into two almost totally separated sedimentary compartments. Their presence is represented by abrupt changes in the orientation of the shoreline approximation in the model. As these changes led to unrealistic values of potential transport, model grid cells #89, 90, and 91 (Fig. 8) are omitted in the further analysis.

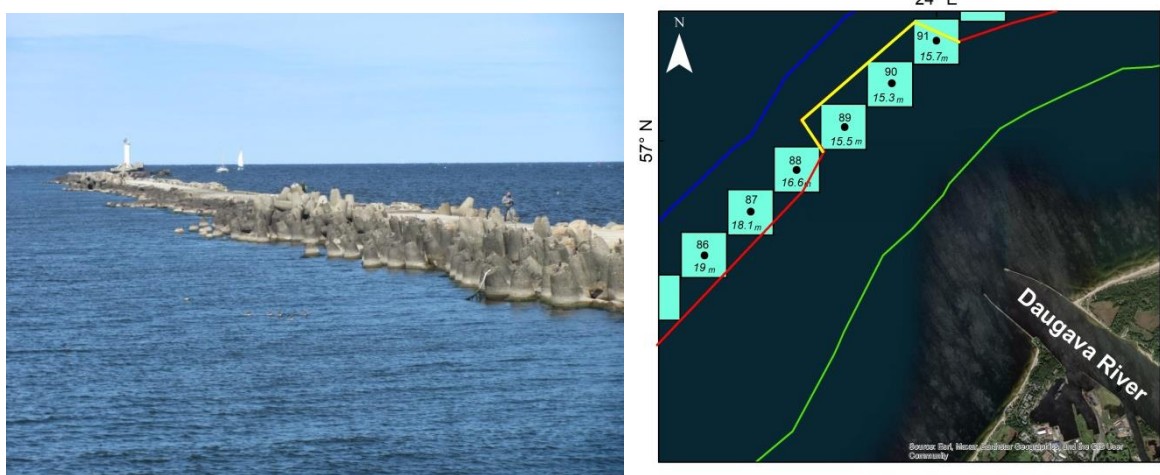

**Figure 8:** Left: Western breakwater at the Daugava River mouth. Photo by T. Soomere, 2019; right: Schematic of the location of wave model grid cells and the approximation of orientation of shoreline (red and yellow) in the transport model.

The local variations in transport are much larger than on the western shore of the Gulf of Riga. The situation between the Port of Engure and Kesterciems (Fig. 9) resembles the situation between Mersrags and Engure (Fig. 7). Both coastal segments contain a few minor headlands that to some extent modulate the intensity of both bulk and net transport, and their ratio. As the orientation of the coastline changes from the north-south alignment at Kesterciems to the almost west-east arrangement at Ragaciems, it is natural that bulk sediment transport slows from the level of about 200,000 m³yr⁻¹ to about 50,000 m³yr⁻¹ in the section between Kesterciems and Ragaciems where waves from the (N)NW approach the shore at a gradually smaller angle. This transport increases again at Ragaciems where both predominant wave systems, one from SW and another from (N)NW (Section 2.2), result in transport in the same direction. It slows down in the vicinity of Jūrmala where northerly waves approach the shore at a small angle and provide only a small contribution to the transport, and waves created by SW winds are weak. The scale of calculations resolves the impact of a small headland at Kauguri (Fig. 4) and the presence of depositional features on both sides of jetties of the Daugava River mouth. The typical bulk sediment transport is from 50,000 m³yr⁻¹ in gently curved coastal segments to 300,000 m³yr⁻¹ near headlands. It is much larger on both sides of the Daugava River mouth and relatively intense (about 150,000 m³yr⁻¹) to the north-east of the Daugava River mouth.

The long-term average transport is predominantly to the south-east and east in the western part of this area, except for single years, such as 2002. Interannual variations in this transport are analysed in Section 3.4. The transport is almost unidirectional (counter-clockwise) in coastal segments to the south of Engure, to the south-east of Ragaciems and in most of the area between mouths of Daugava River and Gauja River (Fig. 9). It is also almost unidirectional along the coast of Jūrmala. The transport direction varies considerably in single years in the area to the north of Ragaciems. The average net transport in single years in coastal segments corresponding to grid cells 21–37 varies from –23,600 m³yr⁻¹ per cell in 2000 to 35,700 m³yr⁻¹ in 1992, with still positive average over all years 10,430 m³yr⁻¹ per cell and 50% of annual values in the range from –13,340 to 8300 m³yr⁻¹. A clear reversal is present near the Daugava River mouth because of a large depositional

feature in this area that modifies the orientation of the coastline, the eastern part of which is being eroded (Bertina et al., 2015).

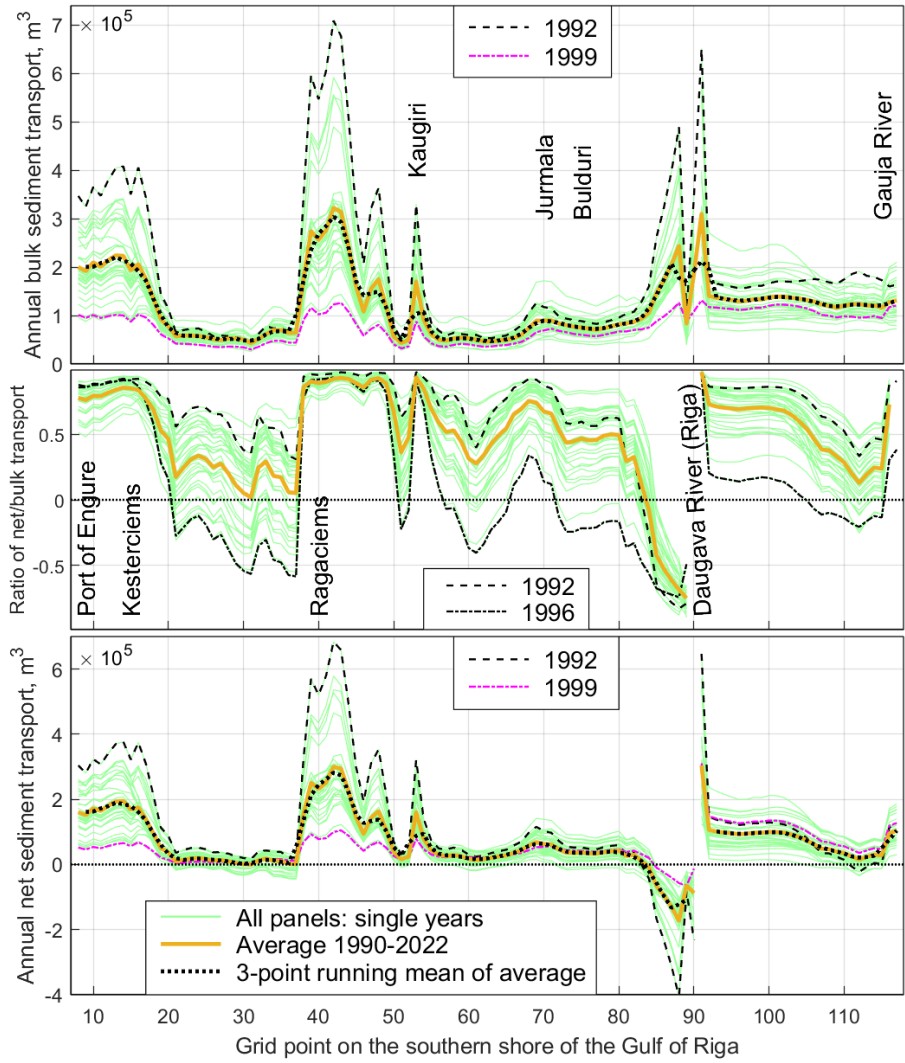

**Figure 9:** Simulated wave-driven potential bulk potential sediment transport (upper panel), ratio of net transport (middle panel) and net transport (lower panel) along the southern shore of the Gulf of Riga from the Port of Engure to the Gauja River mouth. Note different vertical scales of the upper and lower panels compared to Fig 7. The data for grid points that follow the orientation of breakwaters of the Port of Engure and jetties at the Daugava River mouth are omitted (Section 2.5, Fig. 6). See locations in Fig. 4 and a map of the transport scheme in Fig. 13 below.

The average annual net transport is much smaller in this segment, well below 50,000 $m^3yr^{-1}$, with an exception near Engure and around Ragaciems where the simulated average values are almost 200,000 $m^3yr^{-1}$ and up to 600,000 $m^3yr^{-1}$ in single years in a small segment. These estimates match well the historical in situ estimates (Knaps, 1966; Ulsts, 1998); however, earlier lower-resolution simulations for 1970–2007 (Viška and Soomere, 2013b) suggest much more powerful alongshore sediment flux in the vicinity of the Daugava River mouth. Consistent with the above-discussed features,

alongshore net transport is almost zero along the gently curved coastal stretch from Kesterciems to Ragaciems and in the vicinity of Jūrmala. The alongshore variations in transport indicate that the vicinity of Klapkalnciems (where the alongshore net transport decreases, cf Fig. 6) and Jūrmala (the eastern part of which serves as a zero-downcrossing region of net transport, cf Fig. 6) are sediment accumulation areas. A clear reversal of sediment transport at the Daugava River mouth most probably represents the impact of long-term riverine sediment transport into this area since 1567 when the river

established a new entrance into the sea and started to build a new delta (Bertina et al., 2015). This flux of sediment is ignored in the model and only the current geometry of the delta is taken into account. This simplification is appropriate unless the riverine flux is so intense that the added sediment changes the geometry of the shoreline within the study interval. The spatial pattern of net sediment transport signals that wave impact works against the formation of a river delta, consistently with the presence of net sediment flux downcrossing (reflecting a convergence or accumulation area) near the Daugava River

mouth in Fig. 9 and the map of coastline changes 1938–2007: erosion near the southern jetty of the Daugava River and accumulation further to the south until the Lielupe River mouth (Bertina et al., 2015).

Different from the situation on the western coast of the gulf, sediment transport is high along the entire southern coastal stretch in years of intense transport (e.g., 1992) and low along the entire stretch in years of less intense transport (e.g., 1999). The typical correlation coefficients between pointwise values in different years are 0.88 and 0.895 for bulk and net transport,

respectively, with the typical $p$-values <0.0001. The years with intense bulk transport have also strong net transport (e.g., 1992) and vice versa (e.g., 1999). The relevant correlation coefficients between pointwise values of bulk and net transport in single years vary from 0.58 to 0.75 while all $p$-values are $<10^{-11}$. In a similar manner, years with predominantly unidirectional transport have this property along the entire coastal segment (e.g., 1996), except for an approximately 6 km long stretch between the Lielupe River mouth and the western breakwater of the Daugava River mouth while in years with

frequent reversals of this transport reversals occur in about half of this segment. This structure of net transport suggests that the segment in question contains three sedimentary compartments, separated by the headland at Ragaciems and breakwaters of the Daugava River mouth. While sediment from the easternmost system can be transported across the headland at Ragaciems, reverse transport is unusual at an annual scale as the net transport has a zero-upcrossing (and thus a clear divergence point) at this location only in 13 years out of 33 (Fig. 9). The compartment from Kauguri to the western

breakwater of the Daugava River mouth may be considered as a combination of two cells with almost unidirectional sediment exchange between them.

### 3.3 Fragmented eastern shore

The eastern shore of the Gulf of Riga (Fig. 1, 2) from the Gauja River mouth to the Estonian township Häädemeeste (Fig. 4), even though generally almost straight, contains one larger (Cape Kurmrags) and several smaller variations in the coastline

orientation. Historical in situ observations (Knaps, 1966; Ulsts, 1998) suggest that this area may have several erosion and accumulation areas (Fig. 2) and possibly also several sedimentary cells that are more or less isolated from each other in terms of annual sediment transport.

Different from the western and southern shores of the Gulf of Riga, the sandy shore is not continuous in this area. Some coastal segments have cobble and boulder pavement, and consist of material that is not easily erodible, or are rocky (e.g., at Kuiviži). Several coastal segments in the vicinity of the Latvian-Estonian border and Häädemeeste are almost completely devoid of sand and wave driven sediment transport is very limited. Therefore, the actual transport, evaluated in Knaps (1966) and Ulsts (1998), may well be just a small fraction of the simulated potential transport.

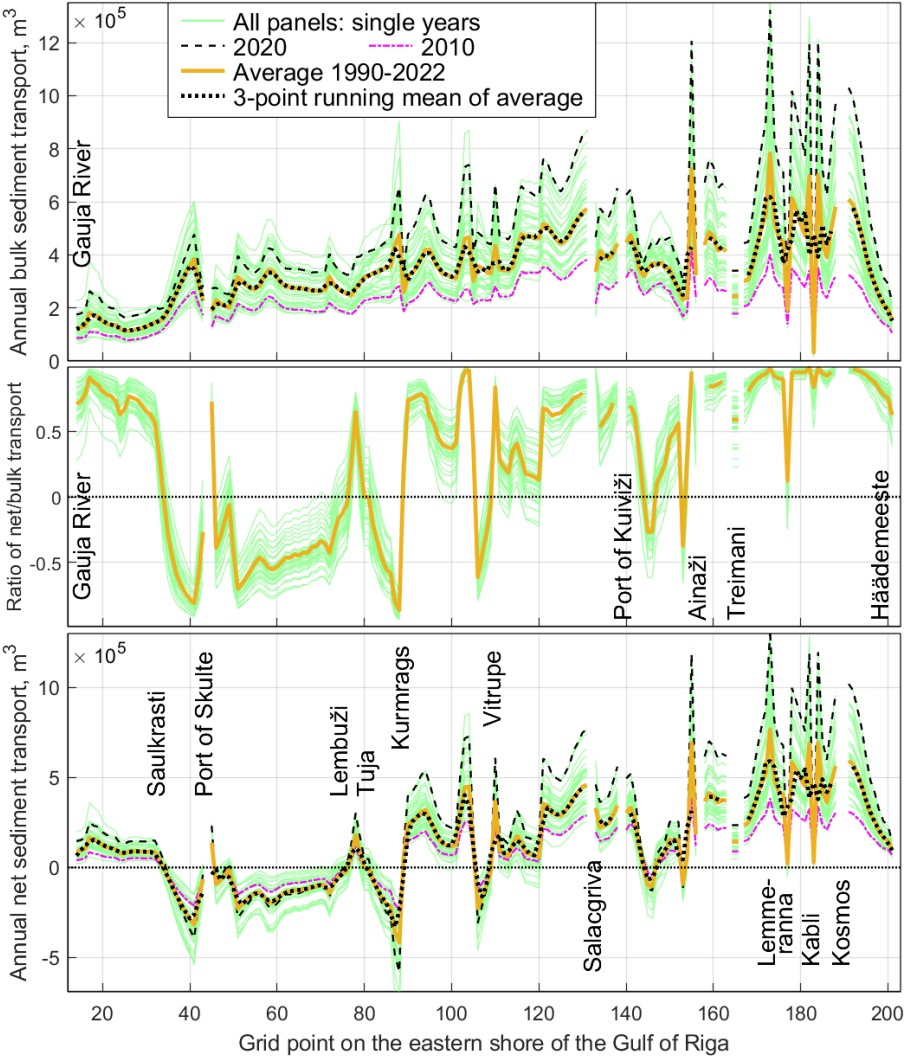

**Figure 10:** Simulated wave-driven potential bulk potential sediment transport (upper panel), ratio of net transport (middle panel) and net transport (lower panel) along the eastern shore of the Gulf of Riga. Note different vertical scales of the upper and lower panels compared to Figs. 7 and 9. The data for grid points that follow the orientation of breakwaters of the Port of Skulte, Salacgriva, Kuiviži, Treimani and at Kosmos are omitted as explained in Section 2.5. See locations in Fig. 4 and a map of the transport scheme in Fig. 13 below.

This coastal segment is evolving, similar to the Latvian and Lithuanian Baltic proper shores, under a delicate balance of two predominant wind and wave systems ((N)NW and SW, Eelsalu et al., 2024b) that in this case work exactly against each

other. This is the natural reason why the potential bulk transport (Fig. 10) increases from about 150,000 m$^3$yr$^{-1}$ in the south to about 400,000 m$^3$yr$^{-1}$ in the north: while the heights of waves generated by the (N)NW winds slowly decrease in this direction because a shorter fetch, the impact of waves excited by SW winds (that is weak in the south of this stretch) considerably increases with the increase of fetch length for these winds.

    The transport direction along this stretch is highly variable (Fig. 10), with typical lengths of stretches of unidirectional
transport of only a few kilometres. The transport in the region immediately to the north-east of the Gauja River mouth is almost fully counter-clockwise to the north-east. The transport is predominantly clockwise from Saulkrasti to Cape Kurmrags, has a variable direction from Cape Kurmrags to Ainaži at the Latvian-Estonian border, and is predominantly to the north (counter-clockwise) in the Estonian part of the study area. This variation apparently mimics changes in the orientation of the shoreline and the changing balance of the fetch lengths of the predominant SW and (N)NW winds. These
lengths are more or less equal in the middle of this coastal stretch. The nearshore of its northern part is to some extent sheltered against waves from the north, north-northwest and north-west by the island of Kihnu and the Estonian mainland.

    Consistent with Viška and Soomere (2013b), the average potential net transport along this stretch varies considerably, between about 15,000 and 590,000 m$^3$yr$^{-1}$ (in terms of 3-point running average, Fig. 10). Its intensity generally increases from the south to the north similar to the bulk transport. There are several persistent zero-upcrossings in the net sediment
transport, together with alongshore variations of the sign of the ratio of net and bulk transport (Fig. 10). These features signal that the sedimentary system of the eastern coast of the Gulf of Riga is highly fragmented. This aspect was not resolved by earlier simulations (Viška and Soomere, 2013b; Soomere and Viška, 2014) that provided a highly generalised picture of the system. Consequently, long-range transport of sediment along this coastal section is unlikely and there are several natural reversals of the overall counter-clockwise sediment transport pattern along with associated sediment erosion and
accumulation regions.

    The presence of several man-made structures, such as the Port of Skulte, jetties at Salacgriva, Kuiviži, Ainaži, Treimani and the historical recreation centre for USSR astronauts (Kosmos in Fig. 10) augments the fragmentation. Together with headlands such as Cape Kurmrags and other smaller headlands that serve as invisible barriers to sediment transport, they separate the coastal stretch into numerous almost isolated sedimentary cells with a typical length of 5–25 km. The longest
interconnected coastal segments are near Saulkrasti (ca 21 km), from the Port of Skulte to Cape Kurmrags (ca 25 km), from Vitrupe to Salacgriva (ca 16 km), and from Treimani to Kosmos (ca 14 km).

    The breakwaters of the Port of Skulte extend to the water depth of about 4 m and the entrance channel to this port is 8–9 m deep (https://www.gpsnauticalcharts.com/main/latvia/lv613310-port-of-skulte-nautical-chart.html). This structure is thus a major obstacle to sediment transport and delineates the northern end of the sedimentary compartment between the port and
the Gauja River mouth. The region to the SW of these jetties apparently is an accumulation area and the area to the north is likely subject to erosion. The accumulation feature at the Gauja River mouth and the associated change in the orientation of the coastline give rise to a local net sediment transport reversal in single years but still allows counter-clockwise sediment flow to the north in most years. A clear sediment flux convergence area at Saulkrasti (Fig. 10) matches the presence of a long

and wide sandy beach. Together with an extensive sediment transport reversal that apparently extends to Lembuži and possibly even to Kurmrags, its presence signals that the Saulkrasti region has been the end location of counter-clockwise sand motion along the rest of the Gulf of Riga shores. This conjecture is supported by the absence of any notable accumulation feature adjacent to the southern breakwater of the Port of Skulte about 6 km to the north of Saulkrasti.

Figure 10 indicates the presence of a persistent reversal area (that is, transport to the south) of net sediment transport to the north of Saulkrasti. This reversal signals that waves from the northern directions dominate the wave-driven transport over this more than 30 km long segment (that is split into two parts by the Port of Skulte). It is not clear whether a minor headland near Lembuži serves as a major barrier of net transport. Even though it creates a zero-upcrossing of annual net sediment transport, the location of this upcrossing varies by several kilometres in single years (Fig. 10). It is thus likely that wave-driven sediment flux passes this headland on many occasions and that the coastal segment from the Port of Skulte to Cape Kurmrags is a connected compartment.

The most significant net sediment flux divergence area is located at Cape Kurmrags, essentially a very minor headland that insignificantly extends into the sea. Together with a sister headland about three kilometres to the north, they are an almost impermeable barrier for wave-driven sediment motion in our model in terms of annual average sediment transport. As single storms still apparently can move sediment around these capes, the sedimentary systems to the north and south of these capes are not totally isolated from each other.

While bulk transport gradually increases from the south to the north between Cape Kurmrags and Salacgriva, net transport greatly varies in this segment. It has a short but clear reversal in terms of annual values near Vitrupe. Similar to the above, it is likely that waves in single storms carry sediment across this location and thus the coastal segment from Cape Kurmrags to Salacgriva is a connected sedimentary compartment. Extensive variations in the intensity of potential net transport indicate areas prone to erosion (if this transport increases from the left to the right, Fig. 6) or accumulation (segments in which the net transport accordingly decreases) in this compartment.

Sediment transport at and to the north of Salacgriva is fragmented. Several minor headlands to the south of Salacgriva modulate the transport properties but do not serve as barriers. Jetties on both sides of the Salaca River and Kuivižu River mouths and especially the >5 m deep entrance channel to Salacgriva almost totally block alongshore sediment transport. The same applies to jetties at Ainaži, Treimani and Kosmos. As the coast to the north of Cape Kurmrags contains very limited fine sediment, the simulated (potential) sediment transport by at least an order of magnitude exceeds the actual wave-driven transport. The nature of the coast and the location and size of accumulation features at different obstacles confirms that the transport is predominantly to the north.

The properties of transport in single years have many particular features in this coastal segment. The years with intense bulk transport generate large transport throughout the segment. In a similar manner, in years with low bulk transport the intensity of bulk transport is low over the entire segment (Fig. 10, upper panel). Interestingly, this feature is not true for the net transport. While its intensity in the northern part of the segment matches the intensity of bulk transport, the situation is

different in the south, especially between the Port of Skulte and Cape Kurmrags, where net transport in these years is at the average level.

### 3.4 Potential bulk and net alongshore sediment transport over the entire area

Estimates of interannual and decadal variations in the bulk and wave-driven potential sediment transport integrated along the eastern Baltic Sea, from Cape Taran to Pärnu Bay, including the western, southern, and eastern coasts of the Gulf of Riga (Soomere et al., 2015) have revealed a major regime shift in transport properties around the year 1990. While potential bulk transport integrated from Samland to Pärnu continued to grow 1970–2007, net transport increased only until about 1990 and decreased 1990–2007. Major changes in the bulk and net transport were clearly visible on the Baltic proper shore of the

Kaliningrad District (of Russia), Lithuania and Latvia but not on the shores of the Gulf of Riga. The bulk potential transport decreased to some extent 1990–2007 on the shores of this gulf while the net transport was at an almost constant level (Viška and Soomere, 2013a).

    A possible reason for the absence of this probable signal of climate change in the Gulf of Riga (Viška and Soomere, 2013a) may be the use of values of potential transport integrated over the entire set of its western, southern and eastern

shores. As these shores are oriented very differently with respect to predominant wind directions from the SW and (N)NW (Section 2.2), it is likely that such a signal is present on some of these shores only.

    The average intensity of potential alongshore sediment transport per grid cell is largest (bulk/net transport about 352,000 / 100,000 $m^3yr^{-1}$) on the eastern shore of the Gulf of Riga (Fig. 2, Fig. 11). The location and orientation of this segment is such that high waves generated by predominant strong SW and (N)NW winds commonly arrive at the coast at a large angle

and thus generate strong alongshore transport. This does not automatically mean massive net or actual sediment transport. Almost the entire eastern shore of the Gulf of Riga (except for an accumulation area at Saulkrasti) suffers from a deficit of sediment (Knaps, 1966; Ulsts, 1998). Consequently, the magnitude of actual sediment transport along this shore is only a small fraction of the potential transport.

    The potential sediment transport is considerably weaker on the other shores of the Gulf of Riga. Its magnitude on the

western shore (bulk/net transport about 226,000 / 54,000 $m^3yr^{-1}$) is, on average, about 64/54 % of that on the eastern shore and only about 34/67 % (bulk/net transport about 119,000 / 67,000 $m^3yr^{-1}$) on the southern shore (Fig. 11). These differences evidently reflect the combination of the direction of predominant SW and (N)NW winds (Section 2.2) and orientation of the coastal segments. While (N)NW winds apparently generate the same magnitude of potential transport on the eastern and western shores, the contribution of waves driven by westerly winds is almost missing on the western shore. This explains the

difference in transport by a factor of two. Similarly, waves created by westerly winds are still low on the southern shore even though they arrive at this shore segment at a large angle. Waves driven by (N)NW winds are commonly much stronger, but they arrive at a small angle and usually do not generate massive alongshore transport.

    The intensity of bulk transport does not increase in the study area (Fig. 11). Different from the properties of this transport integrated over the longer coastal stretch from Samland to Pärnu 1970–2007 (Soomere et al., 2015), this transport decreases

by up to 30% on the eastern shore and in the entire gulf 1990–2005, and exhibits no obvious trend 2005–2022 (Fig. 11). This

pattern is, however, consistent with the course of bulk sediment transport integrated from Cape Kolka to Pärnu Bay in earlier

lower-resolution simulations (Viška and Soomere, 2013a). Interestingly, Viška and Soomere (2013a) also indicated maxima

in this quantity around the years 2004 and 2007.

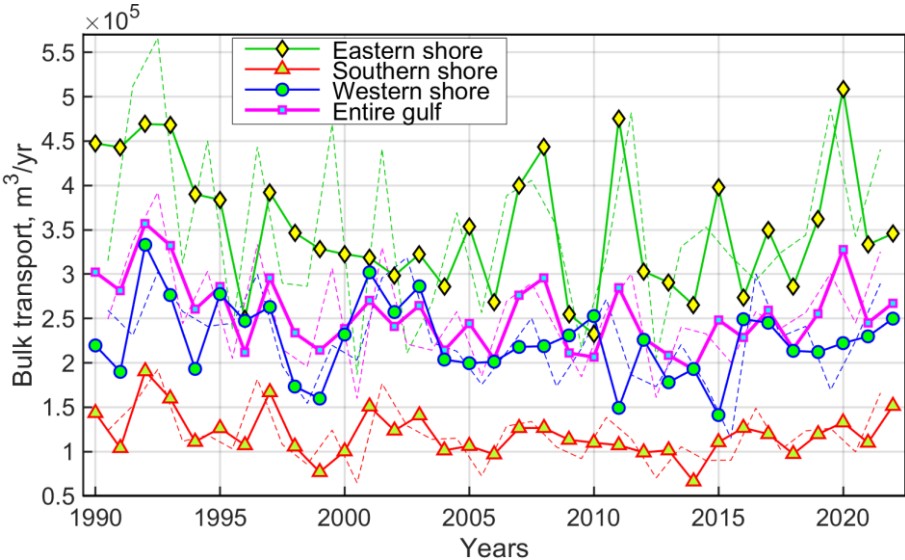

**Figure 11:** Average annual (solid lines with markers) and storm season (thin dashed lines) potential bulk sediment transport per wave model grid cell along western, southern and eastern shores of the Gulf of Riga 1990–2022. The annual bulk transport decreases in 1990–2005 in the entire study area (–5590 m³yr⁻¹ per year) and on the eastern shore (–10,600 m³yr⁻¹ per year), and this decrease is statistically significant ($p = 0.018$ and $p = 0.0008$, respectively). A similar decrease is not statistically significant on the southern and western shore (–920 m³yr⁻¹ per year, $p = 0.75$, and –2160 m³yr⁻¹ per year, $p = 0.20$, respectively). This transport increases slowly (with the relevant 690 slope from 1020 to 1760 m³yr⁻¹ per year) in all addressed coastal segments but this increase is far from being statistically significant as the quantity $p$ is in the range from 0.23 to 0.64.

It is therefore likely that the intensity of wave-driven sediment transport and thus also coastal processes in the interior of

the Gulf of Riga develop independently from (or even in counterphase) with respect to the transport on the shores of eastern

Baltic proper. The probable reason is the presence of long coastal segments in the gulf that are differently oriented with

695 respect to the predominant wind directions from SW and (N)NW.

Another implication of this feature becomes evident as the difference in the pattern of interannual variations of bulk

transport on different shore segments. Namely, transport on the eastern and western shores of the gulf contains extensive

interannual variations (standard deviation (std) 79,000 and 32,000 m³yr⁻¹, respectively) but has no obvious trend (less than

1800 m³yr⁻¹ per year, $p > 0.37$) 2005–2022. The situation was different on the southern shore where transport had large

interannual variations (std 30,000 m³yr⁻¹) in 1990–2005 but has been almost steady (std 18,200 m³yr⁻¹, slow increase by

1025 m³yr⁻¹ per year, $p = 0.23$) since then. It likely that this difference reflects different temporal patterns of changes to

winds from the two predominant directions from SW and (N)NW (Eelsalu et al., 2024b) that become evident differently, in

differently oriented segments.

Additional information about the structure of the temporal course of transport is provided by analysis of transport during so-called storm seasons, specifically, 12-month time periods from July to June of subsequent year (Männikus et al., 2019, Eelsalu et al., 2022). The use of such time periods (Fig. 11, 12) often better characterises the severity of winds in the relatively windy autumn and winter seasons, and thus also of interannual variability of sediment transport intensity. The differences between this quantity and annual bulk transport are relatively large on the eastern shore and fairly small on the southern shore. Consistent with the above, storm season bulk transport does not exhibit any significant trend since 2005. The relevant slopes of the trendlines for the entire gulf and for the western, southern, and eastern shores vary from 1550 to 4300 $m^3yr^{-1}$ per year, with $p$ in the range 0.25 to 0.41.

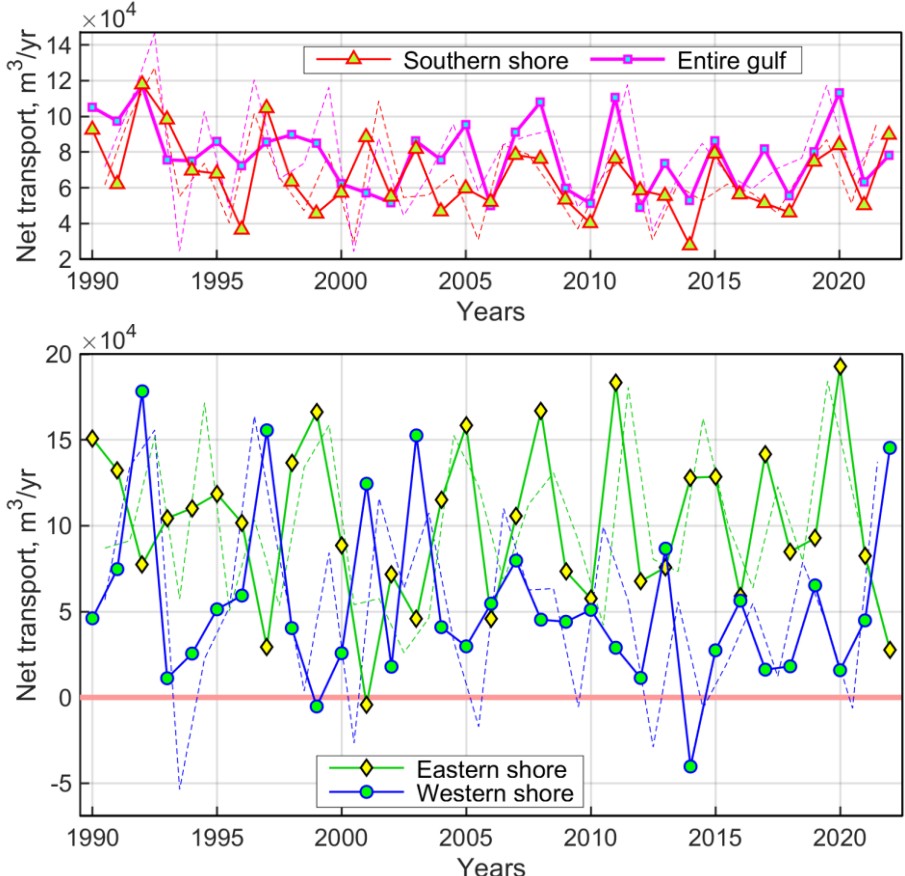

**Figure 12:** Average annual (solid lines) and storm season (thin dashed lines) potential net sediment transport per wave model grid cell along western, southern and eastern shores of the Gulf of Riga in single years 1990–2022.

Different from above but consistent with Viška and Soomere (2013a), average potential net sediment transport integrated over the entire study area (Fig. 12) displays almost no long-term (less than 530 $m^3yr^{-1}$ per year, $p = 0.15$) and decadal changes. It also exhibits much smaller interannual variations than bulk transport in single segments (Fig. 11). Interannual variations in the net transport are, however, significantly different in the three coastal segments. While these variations are fairly limited on the southern shore (std 20,700 $m^3yr^{-1}$), they are much larger on the western and eastern shores (49,000 and

47,330 m$^3$yr$^{-1}$, respectively. While interannual variations in bulk transport are weakly correlated (correlation coefficient 0.19 for annual values and –0.09 for storm season values), interestingly, most of these large variations in net transport are exactly in counter-phase on the western and eastern shores. This feature is less evident in annual values of net transport that have a correlation coefficient –0.16, $p = 0.39$ but impressive and statistically significant at a >99 % level in terms of net transport during storm season, with a correlation coefficient –0.59 and $p = 0.0003$.

The described feature explains why temporal variations in the bulk and net transport integrated over the western, southern, and eastern coasts of the Gulf of Riga are different from those highlighted in Soomere et al., (2015) who identified a gradual increase in the bulk transport over a longer coastal stretch from Cape Taran to Pärnu Bay and a change in the slope of trend in the net transport around the year 1990. The main reason is the presence of the differently oriented eastern shore of the Gulf of Riga. The predominant wind and wave directions from SW and (N)NW act in the same manner in all segments of the stretch from Cape Taran to Pärnu Bay, except for the western shore of the Gulf of Riga. The winds and waves that produce counter-clockwise transport in all other parts of this longer stretch generate clockwise transport on the western shore of the Gulf of Riga (and vice versa) because of its different orientation. When the net transport is integrated over the western and eastern segments of this gulf, these variations cancel each other and lead to limited interannual variations of the total net transport in the entire gulf.

## 4 Discussion and conclusions

The new high-resolution wave data from the SWAN model allowed for a vital update of the earlier estimates of wave-driven potential sediment transport rates, their interannual and decadal variations, the location of divergence and convergence areas of sediment flux, associated patterns of sedimentary compartments and cells on the sedimentary shores of the Gulf of Riga, and further understanding of the difference between some implications of climate change on the Baltic proper shores and in the interior of the Gulf of Riga.

### 4.1 Limitations of simulations

The reliability of estimates of this kind is basically determined by: (i) the quality of input wave information and (ii) limitations of the sediment transport model. The set of wave properties used in our study has been extracted from recent high-resolution simulations of wave fields in the study area using the most contemporary wind information (Section 2.3). The model output has been verified against recorded wave data in many locations of the Baltic Sea (Giudici et al., 2023; Männikus et al., 2024) and the Gulf of Riga (Najafzadeh et al., 2024). Even though the match between reconstructed and recorded wave properties is not always perfect (Eelsalu et al., 2025a), the quality of input wave data is definitely not the main limitation for the quality of the output simulations.

Significantly larger uncertainties are introduced because of the poor resolution of nearshore bathymetry as it affects the wave data. This affects the choice of wave model grid cells (Section 2.4) that are relatively distant from the shoreline in

areas where wave-bottom interaction is relatively weak. The conversion of wave properties in these cells into breaking wave properties (Section 2.4) assumes that the seabed is plane and thus ignores all local features of bathymetry.

The largest differences between simulated and observed transport are introduced by well-known limitations of the CERC model (see Section 2.5). This model only takes into account instantaneous wave properties, assumes unlimited availability of non-cohesive sediment with constant properties in each coastal segment, and ignores cross-shore transport (USACE, 2002). This means inter alia that the result is independent of the actual sequence of storms. Moreover, the CERC model only provides an estimate of potential sediment transport under idealised conditions.

Some other assumptions may contribute to the uncertainties of the model output, as mentioned in Section 2.3. The wave model has been run with an idealised ice-free set-up, the use of which leads to an overestimation of the annual cumulative wave energy flux (Najafzadeh and Soomere, 2024) and thus also bulk transport. Ignoring currents and varying water levels most likely does not substantially affect the results.

## 4.2 General sediment transport patterns

The simulations reinforced the well-known predominant counter-clockwise pattern of wave-driven sediment transport along the western, southern and eastern shores of the Gulf of Riga. The main advance from the material presented here is a more detailed and substantiated pattern of transport, identification of major sediment transit regions and divergence (erosion) and convergence (accumulation) areas on these shores. Together with locations of harbours these areas define the extent and location of the major sedimentary compartments and cells (Fig. 13). The simulations have highlighted different structural properties of sediment transport on the western, southern and eastern shores of the gulf.

The short coastal section immediately to the south-east of Cape Kolka has a clearly visible erosion point associated with a frequent divergence of sediment flux. The western shore from Cape Kolka to a headland to the north of Mersrags has relatively intense counter-clockwise transport that is reversed in some years. It apparently formed a large interconnected sedimentary compartment in the past that is now split into almost isolated cells by breakwaters and jetties. The shore segment to the south of Mersrags to the Port of Engure forms another interconnected sedimentary compartment.

The southern shore has much less intense and more unidirectional counter-clockwise sediment transport that encompasses the entire segment and weakens to the east towards some extensive accumulation areas. The vicinity of the Daugava River mouth became a major end point of this transport after construction of jetties. Part of this transport may have passed the river mouth in the past and reached the ultimate end location at Saulkrasti.

The potential sediment transport is much larger along the eastern shore than the southern shore and increases from the south to the north. This shore contains a longer segment of predominantly clockwise transport and is split into two almost separated sedimentary compartments by an area of divergence of sediment flux near Cape Kurmrags. The compartment to the north of Cape Kurmrags is split into several smaller almost isolated sedimentary cells by breakwaters and jetties. The deficit of fine sediment severely limits the actual transport.

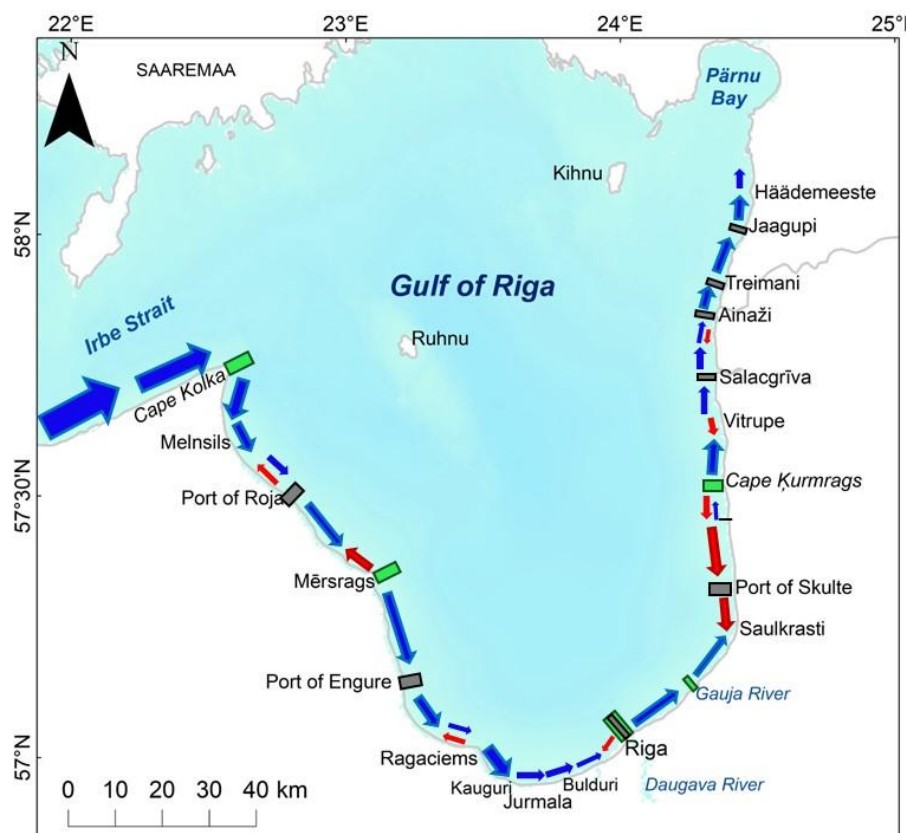

**Figure 13:** Transport directions (arrow widths correspond to the rate of potential net transport), major interconnected sedimentary compartments separated by major natural divergence points of net sediment transport (green rectangles), and large harbours and jetties (black rectangles) that split the sedimentary compartments into almost separated cells. Blue arrows indicate counter-clockwise transport and red arrows show clockwise transport. Parallel narrow blue and red arrows denote variable transport regime in different years.

### 4.3 Interannual and decadal variations in sediment transport

The simulations explained the reason for a mismatch of temporal variations in the wave-driven sediment transport in the interior of the Gulf of Riga in earlier lower-resolution simulations (Viška and Soomere, 2013a) from those identified for longer segments of the eastern Baltic Sea proper (Soomere et al., 2015) as discussed in Section 3.4. The reason is a specific orientation of some shore segments of the gulf with respect to the predominant moderate and strong winds (usually south-western, and north-north-western, Section 2.2, Soomere, 2003) that create the majority of waves responsible for sediment

transport.

These winds generate radically different transport properties on the differently oriented western, southern and eastern shores of the gulf. The western shore is mostly affected by northerly winds. Waves generated by these winds approach the shore at a large angle with respect to the shore normal and thus, if present, drive intense counter-clockwise transport over long distances. Winds from south-west blow to the offshore over this coastal segment and only occasionally contribute to the

clockwise transport. Thus, counter-clockwise transport usually prevails and its magnitude is mostly governed by the properties of the northerly winds.

The southern shore is jointly affected by frequent but relative weak and short waves created by south-western winds and occasionally waves excited by strong but less frequent northerly winds. The latter waves usually approach the shore at a small angle and thus do not generate strong alongshore transport. As a result, the intensity of both bulk and net transport is low and accumulation predominates over long sections of the southern shore.

The eastern shore experiences strong waves generated by both south-western and northerly winds. Both wave systems can be strong and often arrive at the shore at a large angle. Therefore, the direction of transport is jointly covered by these two wave systems. The instantaneous transport direction is thus variable and the annual average reflects the balance of the wave systems in a particular year. The only exception is the southernmost part of the shore at Saulkrasti that is an end point of transport from the west and from the north.

It is therefore natural that the balance of the two components of the local bi-directional structure of moderate and strong winds together with the different orientation of the shoreline in the three coastal segments translates into an interesting mismatch of wave-driven transport properties on the western and eastern shores of the gulf (Fig. 12).

The intensity of bulk transport combines the joint impact of both wave systems and thus largely follows variations in wind speed. The intensity of net transport additionally expresses the changing role of these wave systems. Stronger than average waves from the northerly directions result in stronger than average transport to the south on both eastern and western shores. This means more intense than usual counter-clockwise transport on the western shore and more intense than usual clockwise transport on the eastern shore.

This property naturally translates into a mirrored pattern of time periods of high and low net potential transport on the western and eastern shores of the Gulf of Riga (Section 3.4, Fig. 12). This pattern underscores a highly interesting feature of the dynamics of the Gulf of Riga: almost regular fluctuations in the system with almost constant amplitude and with a time scale of 3–4 years (Fig. 12) that most likely represent the changing the role of northerly winds (Eelsalu et al., 2024b).

## 4.4 Implications for coastal processes

The presented features also translate into several observations with respect to the difference of structural properties of sediment transport and connectivity in the three coastal segments. It is likely that synchronisation of water levels and wave approach (and sediment transport) directions supports the stability of relatively small beaches or sedimentary cells (Eelsalu et al., 2022). This mechanism apparently is not applicable on the western and southern shores of the gulf where large excursions of sediment parcels and long sections of transit are typical. Both these segments contain only one major divergence area that may serve as a barrier for sediment transport and a couple of man-made structures that limit the transport range. This mechanism may, however, become apparent on the eastern shore that is divided into several smaller cells by one major divergence area and several jetties or moles.

The presence of long interconnected sedimentary compartments signals that strong storms may bring large amounts of sediment into motion. A typical consequence of this feature is the rapid straightening of parts of the coast, a process that has already created numerous coastal lakes near the eastern shore and turned the river mouths downdrift on the southern shore of the Gulf of Riga. Another possible consequence is siltation of harbour entrance channels. These processes are much less intense on the eastern shore in spite of the even larger intensity of wave-driven potential transport. A concealed feature is the potential large spread of hazardous materials in the event of sediment contamination along the western and southern shores.

In this context, the presented high-resolution simulations provide valuable insights into sediment transport patterns along the Gulf of Riga coastlines compared to older, essentially basic estimates from in situ observations (Knaps, 1966; Ulsts, 1998) and earlier low-resolution simulations (Soomere and Viška, 2014). These findings aid in the planning of harbour and coastal infrastructure as well in the assessment of several kinds of environmental impacts. It is however not straightforward to link the outcome of our simulations with actual areas of erosion and accumulation (e.g., Luijendijk et al., 2018) because our analysis assumes unlimited availability of fine non-cohesive sediment. Another direct limitation of our study is that it does not take into account cross-shore transport and sediment sources (e.g., from rivers) and sinks.

The decomposition of the sedimentary system of the Gulf of Riga into smaller compartments and cells provides vital information for management solutions and importantly for the identification of potential erosion and accumulation areas. This information is crucial for developing and closing the sediment budget in this microtidal water body. It also indicates how far sediment may be transported from a particular location under the current wind and wave climate. The results are largely invariant with respect to grain size and sediment availability (unless the grain size varies strongly over short distances) even if the potential transport greatly exceeds actual transport. A natural extension of this research would be a similar analysis of sediment transport, compartments and cells along the sedimentary shores of the Baltic proper, ideally including the Polish coastline. Another much-needed extension could be developed using variable locations of the nearshore wave model grid cells. These cells are selected in this study in relatively deep water seaward from the breaker line even in most severe storms. For more often occurring wave conditions the SWAN model is capable of adequately replicating wave properties closer to the shoreline, taking into account wave-bottom interactions that decrease wave energy without generation of massive sediment transport. Such improvements would clearly increase the value of simulation results for users and managers of the coastal area.

*Code and data availability.* Time series of simulated wave properties in the selected wave model grid cells and information about these cells and proxy shoreline orientation are available on request from the authors (mikolaj.jankowski@taltech.ee). The software developed for this study is essentially an almost trivial counting exercise of hourly wave-driven potential transport, and is available on request from the authors.

*Author contribution.* TS and KEP designed the study, created interpretation of the outcome, and prepared the manuscript with contributions from all other co-authors. TS performed analysis of spatial and interannual variations in the transport.

MZJ carried out the analysis of wave data, selection of wave model grid cells and calculations of transport properties, and wrote the relevant sections of the manuscript. ME developed the proxy coastline, validated the outcome and created geographical visualisation. MV contributed geographical and geological data of the study area and linked this information with the outcome of simulations. In the CRediT contributor roles taxonomy: TS: Writing – original draft, review & editing, Visualization, Validation, Software, Methodology, Formal analysis, Supervision, Funding acquisition, Project administration. MZJ: Writing – single parts, Software, Investigation, Validation, Formal analysis, Visualization. ME & MV: Investigation, Validation, Visualization. KEP: Supervision, Methodology, Writing – review & editing.

*Competing interests.* The authors declare that they have no competing interests.

*Acknowledgements.* The research was co-supported by the Estonian Research Council (grant PRG1129) and the European Economic Area (EEA) Financial Mechanism 2014–2021 Baltic Research Programme (grant EMP480). The authors gratefully acknowledge professional and helpful comments of three anonymous referees.

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
