# Peer review of "Alongshore sediment transport analysis for a semi-enclosed basin: a case study of the Gulf of Riga, the Baltic Sea"

_EGUsphere, 2024_

## Referee Comment (RC2)

**Review of** *Alongshore sediment transport analysis for a semi-enclosed basin: a case study of the Gulf of Riga, the Baltic Sea*, **by Soomere et al.**

This study examines the alongshore wave-induced transport of non-cohesive sediment along the shores of the Gulf of Riga. The authors use a triple nested version of the SWAN wave model as input to the CERC approach. They then examine the bulk and net sediment transport along the western, southern and eastern shores of the Gulf of Riga for the period 1990-2022. This study is interesting and clearly provides insights into sediment transport in the area, which is of interest for coastal management issues. However, it still needs some work before it can be published, so I recommend a major revision.

My main comments are as follows:

- The main novel results of the study need to be much better highlighted. Firstly, a more detailed overview of the previous work on the subject and its results and limitations is needed: this will help to highlight the remaining questions addressed in this study and the new insights provided by the present study. Secondly, it is also a question of writing: throughout the manuscript, it is not always clear whether the authors are discussing previous results or their own results; this needs to be made more explicit.

- The tool developed here would allow a much deeper and more quantitative analysis of the interannual variability of sediment transport along the different areas of the coast, both in terms of bulk and net transport, including an analysis of the relationship between this transport and the potential associated factors (variability of the wind regime as well as other human factors). For now, the authors only comment on the 1990-2022 average and some specific years, but they should use their 1990-2022 dataset to perform a more rigorous statistical analysis of interannual variability. For example, calculating the correlation between the time series over different areas, trends and interannual standard deviation, both over the whole period and over selected sub-periods, would allow to be more quantitative than just taking 2 particular years and thus draw much more robust conclusions.
- I would also recommend a more detailed discussion of the associated consequences in terms of erosion or deposition.

- The relationships between sediment transport results and wind and wave characteristics are often presented without illustration or justification. An overview of the wind and wave regime and its spatial and temporal variability over the area is necessary to make these analyses much more sound and convincing. One can also wonder if the resolution of ERA5 is sufficient to correctly represent the spatial and temporal variations of the wind along the coast.

- Some choices in the model configuration need to be discussed or explained. Firstly, only along-shore transport is represented and analysed in the model, what are the implications of neglecting cross-shore transport? The authors mention, for example, storms during which sediment may cross certain structures. Secondly, the effect of man-made or small-scale structures is often mentioned in the manuscript. However, it is not clear how these can be represented in the 600 m resolution model. Thirdly, the relationship between the different grids should be better explained. How do the 3 resolution levels interact and how do the 3 highest resolution grids over the 3 parts of the coast interact.

- The English is generally OK but should be checked, there are some strange sentences with spelling mistakes.

Below are the detailed comments I made when reviewing the paper, many of which are redundant but should help the authors to some extent in responding to my main comments above.

- « The resolution, however, is too low to identify » p. 3 line 70. The resolution of what?
- « the northern and north-eastern parts of this water body » p. 3 line 77. Discuss however the hydrodynamic connexion between this area and the rest of the gulf
- p. 3 Figure 1: Is the red fame in the left panel the 2nd level model area ?
- p. 3 Figure 1 : The colormap in the right panel should be adjusted to make topography easier to visualize
- p. 4) Introduction: Review of existing literature is needed. Before that one would like to have an overview of the existing knowledge and the modeling work done over the gulf of Riga, focused on the gulf or done at larger scale
- « 600 m » p. 4. Resolution of which level ?
- « The presentation follows the classical structure of research papers. » p. 4 line 86. Can be removed
- « study area » p. 4 line 86. An overview of THE study area.
- « 2.1 Study area » p. 4) This part would more relevant in the introduction, with a more detailed analysis of previous work to highlight first the open questions and second the specific novel results of the study in the conclusion.
- « the maximum » p. 4 line 95. Remove « the » before "maximum"
- « The bathymetry » p. 4 line 97. A map with detailed bathymetry would help.
- « The 10 m and 20 m isobaths » p. 4 line 99. Show those isobaths on a map
- p. 4 lines 109113. Which period is covered what by these analysis? What about the temporal ( seasonal to interannual ) variability?
- « temporary reversals » p. 5 line 121: implies temporal variability
- « a specific balance between two predominant wind system » p. 5 line 133. Provide explanations of wind system in the area: spatial and temporal variability
- « of a certain location a » p. 5 line 134. Very vague
- « variation over time » p. 6 line 138. Very vague, some elements are needed to make this argument stronger and be more convincing
- « a triple nested version » p. 6 line 145:
Not clear where the nested grids are from figure 1: are there 3 independent grids within a larger mother grid? from the next paragraph it seems that there are 3 levels but the connection is not clear, as well the connection between the 3 high resolution grids
- « ERA5 wind » p. 6 line 147: is the resolution of ERA5 sufficient ?
- « of currents and varying water levels was ignored » p. 6 line 148: Can this affect the results?
- « a thorough description of the Gulf of Riga wave climate for 1990–2021 » p. 6 line 152: Summarize main results from those study
- « the Gulf of Riga a » p. 6 line 153. Red rectangle in figure 1 right ? This should be indicated in the caption of Figure 1
- « three realisations of a regular rectangular grid » p. 6 line 155: how are the 600 m resolution grids connected? Are they connected with the Gulf of Riga grid ? How are the 3 levels connected? Is this 1-way or 2-way?
- « covered with a similar grid with a resolution of about 300 m » p. 7 line 157. Why? Are simulations performed over this part?
- « sedimen » p. 7 line 163: what about sediment properties ? How are they considered ?moreover this method only applies to non-cohesive sediment this should be explicitly written.
- p. 7) This should be a clear part with a dedicated number 2.3 and title of 2.2 should be adapted accordingly.
- « wave data with 5.5 km resolution data» p. 7 line 174. Source of the data?

- « bathymetry data for proximity to estimated beach closure depth » p. 7 line 179. Reformulate for clarity
- « four-step » p. 7 line 179. Steps 1 and 2 are clear, but please explicitly state steps 3 and 4
- « this » p. 7 line 185. " The " instead of " this"
- p. 8 Figure 4 left: isoline labels are pixelized and can not be read
- « SWAN model simulations 1 » p. 8 Figure 4. Caption. The one with the gulf of Riga grid ?
- « These approximations are not perfect » p. 10 line 432: Is It possible to estimate the associated uncertainty ?
- p. 10 line 244:provide meaning of rhos ( density of sediment), rho (density of water) and p
- « The bulk transport was calculated as the sum of absolute values of ... » p. 10 line 248: the bulk transport is the integral of absolute value of transport over the period right?over the day? OR the whole period of simulation?
- « up to 40 m deep area a few kilometres to the east of Cape Kolka » p. 12 line 297. That would really help to show bathymetric features along the coast here instead of referring to other papers...
- « This deep area becomes evident as a water depth 14–18 m in several selected wave model grid cells located less than 1 km from the shoreline » p. 13 line 298: the sentence seems strange (maybe a word is missing) and it is not clear at all to see to which area in figure 4 the authors refer
- « shoreline (Fig. 4). The 5 m and 10 m isolines meander noticeably 300 between Cape Kolka and Roja. This bottom structure apparently reflects streamlined topographical features in the area stem-ming from Late Weichselian glacial dynamics (Tsyrulnikov et al., 2008 » p. 13 lines 299_301. Same comment:how to visualize those features?
- « in terms of averages » p. 13 line 306: when averaging
- « presence of a small port and its wavebreakers. » p. 13 line 307: how are the port and the wavebreakers represented in the model?
- « shoreline. Still, it is likely that, at least in some years, the overall counter-clockwise sediment transport carries sand 310 around this headland to the south-east.  » p. 13 lines 309.310 : this statement needs to be justified
- « The breakwaters of Port of Engure extend further from the shoreline than those of Port of Roja and Port of Mersrags and apparently discontinue this transport » p. 13 line 310,311: same comment, figure7 does not allow to see this
- « The years characterised by very intense (e.g., 1993) or very low (e.g., 2014) bulk transport along the northwestern shore of Cape Kolka are not mirrored along the coastal stretch to the east of Cape Kolka. » p. 13 lines 317.320:computing correlations between transport west of the cape and east of the cape would be more quantitatively convincing
- « transport. The characteristic feature of the net transport is that years with strong counter-clockwise transport to the west of Cape Kolka (e.g., 2011) correspond to almost zero counter-clockwise transport in the western Gulf of Riga. The change in the sign of the net transport at Cape Kolka in years with strong clockwise transport to the east of the cape (e.g., 2010) » p. 13 lines 321 - 323: again, this statement based on 2 particular years does not allow a robust conclusion. Performing correlations over the whole ensemble of years would be more convincing.
- « evidently reflects the role of northerly winds in such years » p. 13 line 323: without showing maps/ timeseries of wind it is not "evident" at all
- p. 13 Figure 7. It is very interesting that for the ratio and the net transport, maximum year west of the cape becomes minimum at the east, and vice versa.  this should be commented, and again quantitatively checked considering correlations between east and west
- « Interestingly, there is no jump or discontinuity in the average net transport or the ratio of net and bulk transport at this location » p. 13 line 327 - 328: seems contradictory with the previous sentence
- « with previous findings. » p. 13 line 331. Reference is needed

- « It is likely that breakwaters of the Port of Roja largely stop alongshore sediment flux. » p. 14 line 339: same comment as above concerning the port
- « Their impact is not resolved by the model. » p. 14 line 343: again, how could this be considered?
- « even » p. 14 line 346: remove "even"
- « The sandy beach becomes evident again about 10 km to the south of Mersrag » p. 14 line 351. Are there observations that support this sentence?
- « Similar to the described pattern, » p. 14 line 352: what do you mean ?
- « almost fully stop the wave-driven sediment transport. » p. 14) 355 - 357: not visible in figure 7 so how do the authors support this statement?
- « northerly winds while the predominant driver near Riga (Daugava River mouth) and further to the east are south-westerly winds » p. 14 line 361 - 362: a presentation / overview of wind characteristics in the region, and its spatial and seasonal variability, would really help to show the link between the actual wind regime and the wave and shore configuration and the sediment transport
- « The massive breakwaters at the river mouth » p. 15 line 365: again, how are those wavebreakers represented in the model?
- « Port of Engure » p. 15 Figures 7 and 9 suggest a continuity of sediment transport at the port of Engure which is not in agreement with what the authors mention above line 355. This raise the question of how is the connection between the 3 grids dealt with?
- « As the orientation of the coastline changes more to the east at Kesterciems, it is natural that bulk sediment transport slows t » p. 15 line 372-373 and in the whole paragraph: be more specific
- « both predominant wave » p. 15. Same comment
- p. 15 line 369-380. A description of wind and wave regime and direction would help a lot to support affirmations in this paragraph
- « predominantly » p. 15 line 381: this seems true on average and for most of the years but the opposite is observed for some years. This highlights the need to further analyse the interannual variability in a more comprehensive way, based on the analysis of the 1990-2022 set.
- p. 15 Figure 7 and the following ones could be used to better discuss the interannual variability and also to perform statistic analysis over the whole period, to support statements made from the analysis of single years and produce more robust conclusions
- « The data for grid points that follow the orientation of breakwaters of the Port of Engure and jetties at the Daugava River mouth are omitted » p. 16 Figure 9 caption. Not clear.
- « the historical in situ estimates; » p. 16 line 394: reference for these historical estimates?
- « simulations » p. 16 line 394: configuration of those earlier simulations ? how do they differ from the current one?
- « The alongshore variations in transport » p. 16 line 397: Please be more specific mentioning those alongshore variations. Is this the small bump near Jurmala ?
- p. 17. Compute correlations over the years between identified cells would help to better characterize the link between those cells.
- « long-term » p. 17 line 399; how is this riverine sediment flux taken into account in the model?
- « sediment transport is high along the entire coastal stretch in years of intense transport (e.g., 1992) and low along the entire stretch in years of less intense transport (e.g., 1994) » p. 17 line 402: again, using statistical analysis over the whole period would make those conclusions drawn from 2 particular years much more robust.
- « this property along the entire coastal segment » p. 17 line 404: not obvious after jurmala. More generally, the segments east and west of jurmala seem to show distinct behaviors, which seems logical when considering the geographical position of jurmala in figure 4.

- « While sediment from the easternmost system can be transported across the headland at Ragaciems, reverse transport is highly unlikely at an annual scale » p. 17 line 408 - 409: not clear can you explain how you deduce this from figure 9 ?
- « similar to the Latvian and Lithuanian Baltic proper shores, under a delicate balance of two predominant wave systems (Eelsalu et al., 2024b) that in this case work exactly again » p. 17 lines 422-423. Provide details.
- « properties of waves generated by the north-north-western winds » p. 17 line 425: see comments above about wind and waves
- « Figure 10 » p. 18. In figures 7,9,10, what is the meaning of no data?
- « along the eastern shore of the Gulf of Rig » p. 18. For figures 7,9,10 that would be also useful to indicate the distance in km, in addition to points
- « The data for grid points that follow the orientation of breakwaters of the Port of Skulte, Salacgriva, Kuiviži, Treimani and at Kosmos are omitted. » p. 18. Same comment as for figure 7
- « of several man-made structures, » p. 19 line 449 : same comment as above how. How are these human made structures represented in the model at 600m resolution ?
- « waves from the northern directions dominate the wave-driven transport over this more than 30 km long » p. 19 line 466. Same comment as above about wind and waves
- « likely that wave-driven sediment flux passes this headland on many occasions and that the coastal segment from the Port of Skulte to 470 Cape Kurmrags is a connected compartment. » p. 19 line 468-470: how do you support this statement?
- « almost did not become evident on the shores of the 500 Gulf of Riga where bulk potential transport even decreased to some extent 1990–2007 but net transport was at an almost constant level (Viška and Soomere, 2013a). » p. 20 lines 499-501. not very clear.
- « A possible reason …» p. 20 line 502. Do the authors refer to the study of Somere et al 2015 or to this study?
- « of climate change » p. 20 line 503. How is it related to climate change? Wind? Could there be other factors?
- « As these shores are oriented very differently with respect to predominant wind directions, it is likely that such a signal is present on some of these shores only. » p. 20 lines 504-505. Same comment as above about wind and wave regimes and coastline configuration
- « …cell is larges… » p. 20 line 505. Add "In the present study"
- « shore and only about 30 % of that on the southern shore » p. 21 line 512. I guess you mean that transport on the southern shore is about 30% of the one on the western and even less of the eastern?
- « combination of the direction of predominant winds and orientation of the coastal segments » p. 21 lines 513-514. Same comment as above about wind
- « does not increase gradually » p. 21 line 519 . It would help to compute the trend on figure 11: over the whole period and between 1990 and 2005 then 2005 and2022
- « trend 2005–202 » p. 21 line 520. "Over 2005-2022" missing, here and in other places.
- « along western, southern and eastern s » p. 21 Figure 11. What are the limits of those segments?
- « …from, or even in counterphase, with respect to… » p. 21 line 528. To be reformulated
- « interannual variations 1990–2005 but has been almost steady since then » p. 22 line 534. This needs to be more quantitatively and statistically analysed (compute standard deviation, trends, ec...) and factors should be discussed (stormy years?)
- « so-called storm season » p. 22 line 538. Differences and similarities with annual values should be discussed.
- « does not exhibit any significant trend » p. 22 line 542. Needs to be computed
- « course of this quantity signals that the decrease in annual values of the bulk transport along the western shore in the 1990s simply reflects improper clustering of the underlying data into annual values. » p. 22 line 543-544. Not clear.

- p. 22 Figure 12. I recommend to compute correlations between shores to see if they are statistically related
- « long-term and decadal change » p. 22 line 550: should be quantified
- « the three coastal segments. Wh » p. 23 line 552. Western and eastern shoes seem to be negatively correlated in figure 12?
- « vital update … » p. 23 line 558. Needs to be clearly explained: what are the new results here compared to previous studies
- « Figure 13 » p. 24 Figure 13: erosion and accumulation areas should be highlighted.
- « mismatch of decadal variations in the wave-driven sediment transport in the interior of the Gulf of Riga (Viška and Soomere, 2013a) from those identified for longer segments of the eastern Baltic » p. 24 line 589-587. Not clear, should be more explicit
- « the specific orientation of some shore segments of the gulf with respect to the predominant moderate and strong wind » p. 24 line 594-595. Same comment, this needs an overview of wind regime
- « largely follows variations in… » p. 25 line 613. Same comment
- « mirrored pattern of time periods of high and low net potential transport on the western and eastern shores of the Gulf of Riga » p. 25 lines 618-619. Compute correlations to obtain more robust conclusions
- « almost regular fluctuations in the role of northerly winds in the system with almost constant amplitude and with a time scale of 3–4 years» p. 25 line 621 - 621. Not clear

---

## Editor Comment (EC1)

**Review of manuscript egusphere-2024-2640**

Tarmo Soomere, Mikolaj Zbiegniew Jankowski, Maris Eelsalu, Kevin Ellis Parnell, and Maija Viška: Alongshore sediment transport analysis for a semi-enclosed basin: a case study of the Gulf of Riga, the Baltic Sea

**1. General comments**

The authors study coastal sediment transport dynamics in the Gulf of Riga, a semi-enclosed bay at the eastern coast of the Baltic Sea using high-resolution wave time series (SWAN wave model) and the Coastal Engineering Research Centre (CERC) equations for the time span 1990-2022. Based on a hierarchical decomposition of the sedimentary near-coast sedimentary system into compartments and cells, the authors are able to specify transport dynamics along the coast. Integrating these local and regional features the authors draw a generalized picture of the wave-driven potential sediment transport dynamics of the study area. In addition to the natural meteorological, oceanographic and sedimentological environment also anthropogenic coastal structures influencing the transport processes are considered.

The work represents a valuable contribution to the model-based description of coastal dynamics of the eastern Baltic Sea. The methodology can be generalized to be applied to microtidal sandy coastal systems for sustainable coastal zone management.

A publication is recommended after **moderate revision**.

**2. Specific comments**

A restructuring of the text is suggested:

1. The **introduction** should primarily present the scientific task and the concept of its solution based on current knowledge, and quote to references (including authors' publications).

2. A next section should give an overview of the **study area** which is required especially for readers outside the Baltic region who are not familiar with the regional peculiarities. This concerns a description of the geographical, geological, climatic and oceanographic characteristics of the Baltic Sea before the Gulf of Riga is described in more detail, whereby the sedimentological peculiarities of the coast should be taken into account. Sediment sources (including inputs from the open Baltic Sea and discharge from rivers) and sinks should be specified.

The **methods and data** section should include information on all primary (measured) or model-derived secondary data used, as well as a description of the models and their handling (such as the decomposition of the coastal space into cells and compartments and the model grid design). Regarding the models, it concerns the SWAN wave model and the ERA5 model for generating forcing data, the CERC equations and their parameterization.

In a separate section, the **results** already described in the present manuscript should be presented in a coherent manner. However, by now there is a discrepancy between the numerical model approaches and the purely qualitative verbal (or graphic) form of the result descriptions. This discrepancy could be minimized through quantification (increased use of statistically estimated generalizing parameters and parameter functions).

A separate **discussion** section is recommended. In this section, the acceptability of generalized data as model input should be discussed in particular. Other points are the reliability of the results and the limitations of the methods used. It is also important to refer

here to the effects of anthropogenic structures in the coastal area on sediment dynamics, which are mentioned in various places in the text but are not yet discussed sufficiently.

In a final **summary and outlook** section, the results are to be concluded and a perspective is to be given. Figure 13 can be used as a basis for a graphic summary. However, the question arises whether the potential net transport could not be quantified by scaling the corresponding arrows.
In order to facilitate the understanding of the spatial and temporal relationships of the local model results, an additional tabular summary of the results is recommended.
In the outlook, the spatial extension of the investigations already indicated in the last sentence of the manuscript, as well as a methodological refinement for sustainable coastal zone management, should be addressed in more detail.

---

## Author Comment (AC1)

Dear Editor,
Dear Referees,

first of all we would like to thank very much the two referees for professional comments and constructive suggestions. As usual (and is our policy), we have given careful consideration to all general and single comments. Our response is provided using blue italics font.

Referee #1

Thank you for the interesting article. This study provides valuable insights into sediment transport in semi-enclosed water bodies with limited fetch and depths.

*Thank you, we are of course happy with this valuation.*

Although I have only minor comments, I hope they will help make the article easier to read and understand:

1. Introduction

- The terms divergence area (line 34) and sediment flux divergence area (line 48) are not well-defined. For readers unfamiliar with this approach, these terms might be confusing and should be explained more clearly. What do you mean by using these terms? Even if they are explained in detail later, a brief clarification in this section would be helpful.

*Our apologies for being too compact in these locations and forgetting to say that it was divergence/convergence of sediment flux. We have added short explanations: "persistent sediment flux divergence areas which are most likely erosion hotspots" on lines 39–40 of the revised version, and expanded the explanation on lines 53–54 of the revised manuscript (the version with all changes shown; the line numbering may change after the adjustments have been finally integrated).*

- Line 74: It is unclear what the main aim and objectives of the article are. What is the primary purpose of this research? While this is mentioned in Section 4, it should also be highlighted in the Introduction section.

*Thank you. The too compact formulation of the aim and objectives was also highlighted by Referee #2. We have considerably expanded this description (lines 83–90 as well as the insight into the existing results (lines 76–81).*

2. Methods and Data

- Figure 3: Including information about when the picture was taken (spring, autumn, or summer) would be valuable for evaluating the scale of erosion.

*This is a good idea. The photo was taken on 24 August 2013. The trees have fallen several months earlier, most likely during autumn-winter storms 2021/2013 or even earlier as pine needles have evidently dried up some time ago and some vegetation has appeared on the sand.*

- Section 2.2 - **The SWAN model data for the nearshore of the study area**: The explanation of why wave data from the SWAN model is needed begins on line 160. This should be moved to the beginning of the section for better clarity

*Thank you; we have done so. We have also reorganised Section 2, split the material into more subsections and added a description of wind and wave regime in the study are as suggested by Referee #2, so that this section is now Section 2.3.*

- Line 198: The number of grids and their location are shown in Figure 4, not Figure 1.

*Yes, Figure 1 only shows the overall nesting of grids and more detailed information is provided in Figure 4. We have adjusted the text accordingly.*

3. Alongshore Sediment Transport Patterns

- This section might be better renamed Materials and Discussion since, along with presenting the sediment transport pattern analysis, it also include comparisons with previous works. Also part of the section 4 should be moved here, like fig. 13 and it's description.

*We have carefully considered this recommendation but it still seems to us that the original title better represents the content of this section. However, we reviewed the section headings and overall structure of the paper addressing concerns of both referees. See also the comments under Discussion*

*and Conclusions below. To make the flow of thoughts smoother, we have inserted cross-references to Fig. 13 into captions of Figs 7, 9, and 10.*

4. Discussion and Conclusion

- This section should be reshaped into a Conclusion section, as much of the discussion already appears in the previous section.

*We admit that we have, somewhat untraditionally, placed large parts of discussion directly after the relevant results. Our justification is twofold: (1) several results are counter-intuitive and probably needed some comments immediately, (2) properties of transport in different coastal stretches are greatly different, and we decided to help the reader by providing an immediate comparison wherever relevant and necessary. However, we are reluctant to rename this section as it still contains a large portion of overall discussion of presented results.*

- Figure 13 presents one of the main results of this article and should be introduced earlier in the text.

*Thank you; we have inserted early references to this figure in captions to Figs. 7, 9, and 10 and hope that doing so is acceptable with the policy of the journal.*

On behalf of the co-authors
Tarmo Soomere, 02 January 2025

---

## Author Comment (AC2)

Dear Editor,
Dear Referees,

first of all we would like to thank very much the two referees for professional comments and constructive suggestions. As usual (and is our policy), we have given careful consideration to all general and single comments. Our response is provided using blue italics font.

Referee #2
This study examines the alongshore wave-induced transport of non-cohesive sediment along the shores of the Gulf of Riga. The authors use a triple nested version of the SWAN wave model as input to the CERC approach. They then examine the bulk and net sediment transport along the western, southern and eastern shores of the Gulf of Riga for the period 1990-2022. This study is interesting and clearly provides insights into sediment transport in the area, which is of interest for coastal management issues. However, it still needs some work before it can be published, so I recommend a major revision.
*Thank you for this opinion. We admit that there are ways to improve the manuscript, and are happy to incorporate recommendations by the Referee.*

My main comments are as follows:

• The main novel results of the study need to be much better highlighted. Firstly, a more detailed overview of the previous work on the subject and its results and limitations is needed: this will help to highlight the remaining questions addressed in this study and the new insights provided by the present study.
*The problem for us is that there are very few studies of this subject in the past and that all these have been performed by the authors of this manuscript. Three studies have been published a decade ago (Viška and Soomere, 2013a,b; Soomere and Viška, 2014) and one very recently published conference text (Jankowski, M.Z., Soomere, T., Parnell, K.E., Eelsalu, M. 2024. Alongshore sediment transport in the eastern Baltic Sea. Journal of Coastal Research, Special Issue No. 113, 261–265. https://doi.org/10.2112/JCR-SI113-052.1; added to the reference list in the revised version] addresses the similar problem for the western shore of the Kurzeme Peninsula and is thus only conditionally relevant to the current study. We admit that we intentionally left out studies of shoreline relocation in the original submission. This gap is now filled by means of substantial extension of material in Section 2.1.*
*Also, we have now made more clear what is new in our manuscript: the use of considerably increased spatial resolution that allows for the identification of features blocking sediment transport, update of the earlier estimates of wave-driven potential sediment transport rates, their interannual and decadal variations, location of divergence and convergence areas of the sediment flux and associated patterns of sedimentary compartments and cells on the sedimentary shores of the Gulf of Riga, plus understanding why long-term trends in sediment transport in the study area do not match similar trends on the Baltic proper shores*
*These changes have led to substantial rearrangement of the sequence of presentation of material in the Introduction and Section 2, adding factual material into Section 2 and the creation of "new" Section 2.2; however, the line of thoughts and the main arguments have remained the same.*

• Secondly, it is also a question of writing: throughout the manuscript, it is not always clear whether the authors are discussing previous results or their own results; this needs to be made more explicit.
*We admit that we compared all new results immediately with the previous knowledge. As much of the previous results have been formulated by the authors of this manuscript, it probably remained unclear in some locations what exactly is new. We have carefully edited the manuscript for clarity of this aspect.*

• The tool developed here would allow a much deeper and more quantitative analysis of the interannual variability of sediment transport along the different areas of the coast, both in terms of bulk and net transport, including an analysis of the relationship between this transport and the potential associated factors (variability of the wind regime as well as other human factors). For now, the authors only comment on the 1990-2022 average and some specific years, but they should use their 1990-2022 dataset to perform a more rigorous statistical analysis of interannual variability. For example, calculating the correlation between the time series over different areas, trends and interannual standard deviation,

both over the whole period and over selected sub-periods, would allow to be more quantitative than just taking 2 particular years and thus draw much more robust conclusions.

*Thank you for this recommendation. We have added most of the recommended statistical analysis. We only left our more detailed analysis of decadal variability in the study area as it is, to our understanding, not clearly evident and also out of scope of this particular paper.*

- I would also recommend a more detailed discussion of the associated consequences in terms of erosion or deposition.

*We agree that discussion of these items is important; however, as we evaluate the magnitude and direction of **potential** sediment transport and do not address other important features, such as availability of sediment, cross-shore transport or sediment flux from rivers, discussion of this kind would remain purely theoretical and might even lead to misinterpretations. For this reason we think it is justified to limit our discussion to the impact of major natural divergence and convergence regions of wave-driven sediment flux and to a couple of cases where spatial variations in the evaluated alongshore sediment present clear indication about erosion or accretion, for example, the vicinity of breakwaters at the Daugava River mouth.*

- The relationships between sediment transport results and wind and wave characteristics are often presented without illustration or justification.

*We agree with this observation; however, our excuse is that (1) wave properties mostly follow local wind properties in this almost inland water body of mostly regular shape and (2) we rely on the CERC formula that relates wave characteristics with sediment transport. To make this position explicit, we have inserted a longer description of wind and wave climates and their interrelations into Section 2 as "new" Section 2.2.*

- An overview of the wind and wave regime and its spatial and temporal variability over the area is necessary to make these analyses much more sound and convincing.

*Thank you for this comment. We definitely agree. Our excuse for omitting this material in the original submission was that most of the relevant studies were performed by the authors of this manuscript. As we did not wish to include many our own papers into the list of references, we only included a very limited description of properties of the local wind and wave regime into the original submission. Having said this, we are happy to expand this description so that the reader does need to consult with other sources. Thus, we have included such overview, including an insight into some specific features of interrelations between wind and wave-driven transport regimes in (the vicinity of) the study area that have been identified most recently, as "new" Section 2.2 into Section 2.*

- One can also wonder if the resolution of ERA5 is sufficient to correctly represent the spatial and temporal variations of the wind along the coast.

*As the overwhelming majority of properties of waves that impact coastal sediment is driven by winds in the open part of the sea, it is more important to be convinced that offshore wind properties are adequately represented by the wave model and its driving wind forcing. We have analysed this aspect in two publications about wave modelling in the Gulf of Finland (Giudici et al., 2023) and Gulf of Riga (Najafzdeh et al., 2024), and also asked the same question specifically for the Gulf of Finland [Männikus, R., Soomere, T., Suursaar, Ü. 2024. How do simple wave models perform compared with sophisticated models and measurements in the Gulf of Finland? Estonian Journal of Earth Sciences, 73(2), 98–111, https://doi.org/10.3176/earth.2024.10; added to the list of references]. The answer is that the wave model we use and ERA5 forcing provide adequate representation of wave properties. Another problem is that spatial resolution of the existing bathymetry data of the nearshore of the Gulf of Riga is insufficient in many locations. For this reason it does not make sense to run our simulations at an even higher resolution.*

- Some choices in the model configuration need to be discussed or explained. Firstly, only along-shore transport is represented and analysed in the model, what are the implications of neglecting cross-shore transport? The authors mention, for example, storms during which sediment may cross certain structures. Secondly, the effect of man-made or small-scale structures is often mentioned in the manuscript. However, it is not clear how these can be represented in the 600 m resolution model. Thirdly, the relationship between the different grids should be better explained. How do the 3 resolution levels interact and how do the 3 highest resolution grids over the 3 parts of the coast interact.

*Thank you for addressing these items. (1) Yes, we only consider alongshore transport. The main reason is that alongshore transport is exceptionally intense in the study area because waves often approach the shore at a large angle. As a natural consequence, cross-shore transport plays usually much smaller role, except for a few locations (e.g., Šakurova et al., 2025). The core implication of neglecting cross-shore transport is that shoreline relocation may have a different rate compared to the case when cross-shore transport is taken into account. However, our conclusions only concern distribution of convergence and divergence regions and the impact of man-made structures to the alongshore transport, and thus are invariant with respect to the impact of cross-shore transport. We have added the relevant explanation into the revised version. Our remark about storms that can transport sediment across some structures has only in mind situations when alongshore transport is created at larger depths than the depth down to which a coastal structure (jetty or harbour) blocks alongshore transport. (2) We have added a detailed explanation how the major man-made structures are represented in the model into "new" Section 2.5. (3) We have also added a more detailed explanation how the nested grid structure works into "new" Section 2.3.*

• The English is generally OK but should be checked, there are some strange sentences with spelling mistakes.

*Thank you; we have carefully language-edited the revised version of the manuscript.*

Below are the detailed comments I made when reviewing the paper, many of which are redundant but should help the authors to some extent in responding to my main comments above.

*Thank you for all these observation. We have carefully considered all these items. As several comments address the same gap in the original submission, we repeatedly refer to the same upgrade.*

• «The resolution, however, is too low to identify» p. 3 line 70. The resolution of what?

*The spatial resolution of the transport model used in Soomere and Viška (2014) is meant as explained in the revised version, line 85.*

• «the northern and north-eastern parts of this water body» p. 3 line 77. Discuss however the hydrodynamic connexion between this area and the rest of the gulf

*This area is included into the wave generation area of the wave model in use. We have included the relevant remark into this location on lines 190–191 of the revised version.*

• p. 3 Figure 1: Is the red frame in the left panel the 2nd level model area?

*Yes, it is the area covered by the 2$^{nd}$ level wave model grid, to properly represent the wave fields entering the Gulf of Riga directly from the Baltic proper and via the West Estonian Archipelago. We added the relevant comment to the caption of Figure 1. The entire left panel represents the area covered by the outermost grid and the red box represents the area covered by the 2nd level wave model grid. The size of the 2nd level rid is chosen to properly represent the wave fields entering the Gulf of Riga directly from the Baltic proper and via the West Estonian Archipelago.*

• p. 3 Figure 1: The colormap in the right panel should be adjusted to make topography easier to visualize

*Thank you; we have done so. We have also added 10 m and 20 m isobaths.*

• p. 4) Introduction: Review of existing literature is needed. Before that one would like to have an overview of the existing knowledge and the modeling work done over the gulf of Riga, focused on the gulf or done at larger scale

*We fully agree and have provided this kind of review in the revised manuscript. The "problem" here is that the entire pool of papers on this topic in international research literature consists of three publications of the authors of the current manuscript: (Viška and Soomere, 2013a,b; Soomere and Viška, 2014). Another work by authors of this manuscript (Soomere et al., 2015) looks at the simulations in the mentioned papers from another (climate change) viewpoint. As this viewpoint is important, we have expanded the presentation in Introduction to cover its output as well. All other sources are derivatives of these four publications.*

*A more detailed description of observations of coastal processes in the study area is still accommodated in Section 2.1 "Study area" where it seems (at least to our eyes) better placed. This description has been*

*substantially expanded to cover all available results about relocation of coastline in the study area from Cape Kolka to Pärnu Bay in international research literature. Together, these two sets of evidence provide comprehensive overview of all modelling and observation efforts of coastal changes in the study area.*

• «600m » p. 4. Resolution of which level ?

*The resolution of the three innermost grids is meant. We have added this information into the text, line 193 of the revised manuscript.*

• «The presentation follows the classical structure of research papers.» p. 4 line 86. Can be removed

*Yes, we agree, and have done so.*

• «study area» p. 4 line 86. An overview of THE study area.

*Yes, indeed, thank you.*

• « 2.1 Study area » p. 4) This part would more relevant in the introduction, with a more detailed analysis of previous work to highlight first the open questions and second the specific novel results of the study in the conclusion.

*We only partially agree here with the Referee in the sense that a part of the material in Section 2.1 might well be placed in Introduction. However, it is customary to place the description of study area into a separate subsection somewhere after Introduction. Thus, we have left part of the material of Section 2.1 where it was, and have amended it with facts about relocation of the shoreline as described above. Similar to above, the "problem" is that there are very few previous studies/observations of properties of alongshore sediment transport.*

*Also, we have considerably expanded the formulation of the core research questions as described in the response to main question 1. However, we feel that we have clearly formulated the specific novel results at the very beginning of Section 4, so this part did not need substantial changes.*

• « the maximum » p. 4 line 95. Remove « the » before "maximum"

*'The' has been removed as suggested.*

• « The bathymetry » p. 4 line 97. A map with detailed bathymetry would help.

*We redressed Figure 1 so that it shows more clearly the basic features of bathymetry that are essential for our study, such as regular appearance of bathymetry in the study area, a deep region near Cape Kolka, and the basic shape of 10 and 20 m isobaths. We still think that it is useful to refer to professional maps in open-source papers, such as Tsyrilnikov et al. (2008, 2012).*

• «The 10 m and 20 m isobaths» p. 4 line 99. Show those isobaths on a map

*These lines are added to Figure 1.*

• p. 4 lines 109–113. Which period is covered what by these analysis? What about the temporal (seasonal to interannual) variability?

*The older papers, such as Knaps (1966) are very compact and do not explain in detail which period is covered. To make this clear, we say "**apparently** stemming from the 1960s and updated in the 1990s" in the revised manuscript. No additional information is provided in later Latvian-language publications.*

*We checked also all available Latvian-language publications. Even though they often mention that most processes are happening during stormy season, we did not find any consequent seasonal or interannual variability analysis.*

• «temporary reversals» p. 5 line 121: implies temporal variability

Yes, it does, and this is exactly what we mean, having in mind the Oxford dictionary meaning "for a limited period of time; not permanently": the reversals appear in a few single years. As this feature is explained in detail in an open source publication (Viška and Soomere, 2013b), we prefer to leave this explanation as is.

• «a specific balance between two predominant wind system» p. 5 line 133. Provide explanations of wind system in the area: spatial and temporal variability

*This sentence has been rearranged to express our point differently. As described above, we have added a more detailed description of the specific wind system in the study area and an insight of the consequences of the presence of such a system on sediment transport properties and coastal equilibrium into new*

*subsection 2.2 so that the notion of "balance" of the impact of two wave systems is first mentioned at the very end of subsection 2.2.*

• «of a certain location a» p. 5 line 134. Very vague

*This item is now described in the Introduction. We tried to better explain that this location is a function of the balance of the impact of two competing wave systems.*

• «variation over time» p. 6 line 138. Very vague, some elements are needed to make this argument stronger and be more convincing

*We have added the estimate of Viška and Soomere (2013b): four grid points or >20 km.*

• «a triple nested version» p. 6 line 145: Not clear where the nested grids are from figure 1: are there 3 independent grids within a larger mother grid? from the next paragraph it seems that there are 3 levels but the connection is not clear, as well the connection between the 3 high resolution grids

*The explanation of the grid system is provided in the next paragraph. We have also added the relevant explanation to the caption of Figure 1.*

• «ERA5 wind» p. 6 line 147: is the resolution of ERA5 sufficient?

*We think it is, and have expressed our arguments in response to the main comments.*

• «of currents and varying water levels was ignored» p. 6 line 148: Can this affect the results?

*Varying water levels do not affect the results because we only consider the idealised case of potential transport that is independent of the particular water level. The main limiters of the accuracy of calculations are the quality of wind and bathymetry information. The presence of currents may modify wave properties to some extent but there is currently no way to reliably replicate the current system of the Gulf of Riga. The ignoring of ice cover apparently leads to an overestimation of transport up to 20% (Najafzadeh and Soomere 2023).*

• «a thorough description of the Gulf of Riga wave climate for 1990–2021» p. 6 line 152: Summarize main results from those study

*Yes, this is done in new Section 2.2, so we inserted a reference to this section into this location.*

• «the Gulf of Riga a» p. 6 line 153. Red rectangle in figure 1 right? This should be indicated in the caption of Figure 1

*Thank you, this has been added. We have also added the remark that the 1 nmi resolution grid also covers the vicinity of the Gulf of Riga.*

• «three realisations of a regular rectangular grid» p. 6 line 155: how are the 600 m resolution grids connected? Are they connected with the Gulf of Riga grid ? How are the 3 levels connected? Is this 1-way or 2-way?

*Thank you for raising this question. The nesting is organised in 1-way manner between all levels. We do not care about the impact of waves generated in the Gulf of Riga on wave fields in the open Baltic proper. Also, the properties of waves generated in 600 m grids during winds that blow from the shore to the offshore are basically the same as those generated in the 1 nmi grid because small water depths (that are replicated better in the 600 m grids) do not really affect wave generation properties. Theoretically, the difference reflects the difference of fetch length that could be by 600 m longer for 600 m grids – but this difference is unimportant over 10s of grid points. The 600 m grids get information about wave properties at their borders from the 1 nmi grid. The three 600 m grids are only connected with the 1 nmi grid and not between each other. We have included a short insight into the grid structure into (new) section 2.3 (formerly 2.2).*

• «covered with a similar grid with a resolution of about 300 m» p. 7 line 157. Why? Are simulations performed over this part?

*This is because of more complicated bathymetry and coastal geometry in this part of the Gulf of Riga. As Wave simulations performed over this part are not used in our study, this remark has been deleted.*

• «sediment» p. 7 line 163: what about sediment properties ? How are they considered? Moreover this method only applies to non-cohesive sediment this should be explicitly written

*We assume constant sediment properties along the entire study area. This is of course not a perfect assumption but allows a comparison of our results with the outcome of earlier studies. The assumed*

*properties of sediment are specified in (new) Section 2.5 (previous 2.4). Thank you for reminding us that sediment is assumed to be non-cohesive (and we have made this clear).*

- p. 7) This should be a clear part with a dedicated number 2.3 and title of 2.2 should be adapted accordingly.

*Thank you for this recommendation. However, our feeling is that this segment is a coherent part of the presentation that should be kept in one unit even though this unit is somewhat longer than previous subsections.*

- «wave data with 5.5 km resolution data» p. 7 line 174. Source of the data?

*The data sources (Räämet and Soomere 2010; Soomere and Räämet, 2011, 2014) were indicated in (Soomere et al., 2017). As these sources are the own publications of one of the authors, we were reluctant to include them into the initial version of the manuscript but we are happy to do so in the revised version.*

- «bathymetry data for proximity to estimated beach closure depth» p. 7 line 179. Reformulate for clarity

*Yes, it was clumsy indeed and has been adjusted.*

- « four-step » p. 7 line 179. Steps 1 and 2 are clear, but please explicitly state steps 3 and 4.

*Thank you, steps 3 and 4 have been formulated in the revised version.*

- «this» p. 7 line 185. " The "instead of" this"

Thank you; we have reshaped the entire sentence on lines 182–185.

- p. 8 Figure 4 left: isoline labels are pixelized and can not be read

*Thank you; we shall provide a higher-resolution image for publication.*

- «SWAN model simulations 1» p. 8 Figure 4. Caption. The one with the gulf of Riga grid?

*Yes, thank you; the 1 nmi grid that covers the Gulf of Riga and its vicinity. We have made this clear in the revised version.*

- «These approximations are not perfect» p. 10 line232: Is It possible to estimate the associated uncertainty?

*Not really. Many people have addressed this issue but the results are strongly site-specific. This is why we presented many references.*

- p. 10 line 244:provide meaning of rhos (density of sediment), rho (density of water) and p

*Yes, indeed.*

- « The bulk transport was calculated as the sum of absolute values of ... » p. 10 line 248: the bulk transport is the integral of absolute value of transport over the period right?over the day? OR the whole period of simulation?

*Yes, exactly. We used different time periods, from months to the whole period of simulation, and added this interpretation into the revised version.*

- «up to 40 m deep area a few kilometres to the east of Cape Kolka» p. 12 line 297. That would really help to show bathymetric features along the coast here instead of referring to other papers...

*Thank you; we adjusted Figure 1 so that this feature is visible. As the manuscript is already fairly long and high-resolution bathymetry images are available in many sources, we are reluctant with respect to adding more material and prefer, if acceptable, to refer to other existing resources.*

- «This deep area becomes evident as a water depth 14–18 m in several selected wave model grid cells located less than 1 km from the shoreline» p. 13 line 298: the sentence seems strange (maybe a word is missing) and it is not clear at all to see to which area in figure 4 the authors refer

*Thank you, the word "selected" was excessive. Also, there must be "Fig. 5, left panel", our apologies.*

- shoreline (Fig. 4). The 5 m and 10 m isolines meander noticeably 300 between Cape Kolka and Roja. This bottom structure apparently reflects streamlined topographical features in the area stemming from Late Weichselian glacial dynamics (Tsyrulnikov et al., 2008)» p. 13 lines 299_301. Same comment: how to visualize those features?

*Visualisation of these small-scale features needs adding one more map to the manuscript. As the manuscript is already fairly long and high-resolution bathymetry images are available in many sources, we*

*prefer, if acceptable, to refer to other sources that have specifically highlighted streamlined topographical features in the area (Tsyrulnikov et al., 2008).*

• «in terms of averages» p. 13 line 306: when averaging
*Yes, thank you.*

• «presence of a small port and its wavebreakers.» p. 13 line 307: how are the port and the wavebreakers represented in the model?
*A longer explanation of this aspect has been added to "new" Section 2.5, with a scheme of the location model grid points near the Port of Roja. Shortly: these ports/wavebreakers stop wave-driven sediment flux.*

• «shoreline. Still, it is likely that, at least in some years, the overall counter-clockwise sediment transport carries sand around this headland to the south-east.» p. 13 lines 309–310 : this statement needs to be justified
*Yes, we have added the remark: "[because] the values of net transport are positive in some years along the entire shore of this headland (Fig. 7)". This is even better visible from positive values of the ratio of net and bulk transport.*

• «The breakwaters of Port of Engure extend further from the shoreline than those of Port of Roja and Port of Mersrags and apparently discontinue this transport» p. 13 line 310, 311: same comment, figure 7 does not allow to see this
*Yes, we have formulated the reason: "The breakwaters of this port extend even further from the shoreline than those of Port of Roja and Port of Mersrags into clearly deeper water than closure depth (>3.5 m in this location) and apparently discontinue wave-driven alongshore transport" as explained in Section 2.5.*

• «The years characterised by very intense (e.g., 1993) or very low (e.g., 2014) bulk transport along the northwestern shore of Cape Kolka are not mirrored along the coastal stretch to the east of Cape Kolka.» p. 13 lines 317–320: computing correlations between transport west of the cape and east of the cape would be more quantitatively convincing
*Yes, indeed, the correlation coefficient of bulk transport over all 22 grid points to the west of this cape and 22 points to the east of this cape is 0.016. We added this information to the text.*

• «transport. The characteristic feature of the net transport is that years with strong counter-clockwise transport to the west of Cape Kolka (e.g., 2011) correspond to almost zero counter-clockwise transport in the western Gulf of Riga. The change in the sign of the net transport at Cape Kolka in years with strong clockwise transport to the east of the cape (e.g., 2010)» p. 13 lines 321 - 323: again, this statement based on 2 particular years does not allow a robust conclusion. Performing correlations over the whole ensemble of years would be more convincing.
*We agree. The similar correlation coefficient as above, but for net transport in single years, is –0.6125, with p=0.0002 indicating statistically significant negative correlation between these values.*

• «evidently reflects the role of northerly winds in such years» p. 13 line 323: without showing maps/ timeseries of wind it is not "evident" at all
*Here we do not agree as strong clockwise (=to the south on western shore of Cape Kolka) transport can be only driven by waves that approach from the northern directions and thus are generated by northerly winds. However, we have amended the wording so that this aspect is made explicit. We also corrected one interpretation error.*

• p. 13 Figure 7. It is very interesting that for the ratio and the net transport, maximum year west of the cape becomes minimum at the east, and vice versa. this should be commented, and again quantitatively checked considering correlations between east and west
*This is interesting indeed but, in essence, a straightforward outcome of the orientation of the coastline at different sides of Cape Kolka. We have explained this feature in more detail in the revised version.*

• «Interestingly, there is no jump or discontinuity in the average net transport or the ratio of net and bulk transport at this location» p. 13 line 327-328: seems contradictory with the previous sentence
*Thank you; the claim was totally wrong by some strange reason, possibly because of unwanted copy/paste of some phrases. In fact, there is no jump or discontinuity in the average bulk transport. The text has been corrected.*

- «with previous findings.» p. 13 line 331. Reference is needed

Yes, we agree, and inserted the necessary references.

- «It is likely that breakwaters of the Port of Roja largely stop alongshore sediment flux.» p. 14 line 339: same comment as above concerning the port

*We agree that this aspect needs more in-depth explanation. We have done so by extending Section 2.5 towards explaining that wave-driven sediment transport mostly occurs in the nearshore, from the shoreline down to closure depth, and that structures that extend even deeper water almost entirely stop this transport. Based on this explanation, we explain in the revised version how deep is the entrance channel of harbours and how far/deep the harbour breakwaters or jetties extend.*

- «Their impact is not resolved by the model.» p. 14 line 343: again, how could this be considered?

*We explain how the model works near harbours in more detail in Section 2.5. The presence of these small structures is not reflected in the model set-up. We have adjusted the text to make this aspect clear.*

- «even» p. 14 line 346: remove "even"

*Here we respectfully disagree as we wish to underline that the existing separation of the shore into sedimentary compartments contain several features that are not man-made.*

- «The sandy beach becomes evident again about 10 km to the south of Mersrags» p. 14 line 351. Are there observations that support this sentence?

*Yes, this is visible from Google Maps and Google Earth.*

- «Similar to the described pattern,» p. 14 line 352: what do you mean?

*These words are removed.*

- «almost fully stop the wave-driven sediment transport.» p. 14) 355 - 357: not visible in figure 7 so how do the authors support this statement?

*As explained above, we refer to the material in Section 2, extension of breakwaters, depth on the entrance channel and closure depth.*

- «northerly winds while the predominant driver near Riga (Daugava River mouth) and further to the east are south-westerly winds» p. 14 line 361 - 362: a presentation / overview of wind characteristics in the region, and its spatial and seasonal variability, would really help to show the link between the actual wind regime and the wave and shore configuration and the sediment transport

*Thank you; we have provided an overview of the wind regime and some features of its spatial variability in the Gulf of Riga in (new) Section 2.2. As we are not interested in seasonal variability in transport, we have not presented the relevant information about the local wind climate.*

- «The massive breakwaters at the river mouth» p. 15 line 365: again, how are those wavebreakers represented in the model?

*We have included the relevant schematic into Fig. 8 and explained at the beginning of Section 3.2 that their presence is represented by abrupt changes in the orientation of the shoreline approximation in the model. As these changes led to unrealistic values of potential transport, values of this transport in model grid cells #89, 90, and 91 (Fig. 8) are omitted in the further analysis.*

- «Port of Engure» p. 15 Figures 7 and 9 suggest a continuity of sediment transport at the port of Engure which is not in agreement with what the authors mention above line 355. This raise the question of how is the connection between the 3 grids dealt with?

*Thank you; we explain now more clearly that unrealistic values of sediment transport are left out from images and from the analysis. We also explain in more detail where sediment flux has discontinuities. As this description for Engure has been moved to the very end of the previous section, it seems not necessary to mention that again.*

- «As the orientation of the coastline changes more to the east at Kesterciems, it is natural that bulk sediment transport slows» p. 15 line 372-373 and in the whole paragraph: be more specific

*This was too short indeed. We say in the revised version that "the orientation of the coastline changes from the north-south alignment at Kesterciems to the almost west-east arrangement at Ragaciems".*

- «both predominant wave» p. 15. Same comment

*To make things clear, we have added explanation "one from SW and another from (N)NW" and also a cross-reference to (new) Section 2.2.*

- p. 15 line 369-380. A description of wind and wave regime and direction would help a lot to support affirmations in this paragraph.

*We agree and have included the relevant explanation to Section 2 ((new) Section 2.2).*

- «predominantly» p. 15 line 381: this seems true on average and for most of the years but the opposite is observed for some years. This highlights the need to further analyse the interannual variability in a more comprehensive way, based on the analysis of the 1990-2022 set.

*Thank you. As this analysis is provided in the next section (3.4), it was indeed necessary to add the reference to this section. We have added the remark "except for single years, such as 2002" as the average net transport was negative only in this year.*

- p. 15 Figure 7 and the following ones could be used to better discuss the interannual variability and also to perform statistic analysis over the whole period, to support statements made from the analysis of single years and produce more robust conclusions

*Yes, thank you; we have provided this analysis in Section 3.4 for all three large segments. We have also included several aspects of local transport statistics into Sections 3.1, 3.2, and 3.3 of the revised version.*

- «The data for grid points that follow the orientation of breakwaters of the Port of Engure and jetties at the Daugava River mouth are omitted» p. 16 Figure 9 caption. Not clear.

*We have included references to amended Section 2.5 and to the added panel in Fig. 6 that explain the situation.*

- «the historical in situ estimates;» p. 16 line 394: reference for these historical estimates?

*The references (Knaps, 1966; Ulsts, 1998) have been added.*

- «simulations» p. 16 line 394: configuration of those earlier simulations ? how do they differ from the current one?

*They had much lower resolution and covered a different time interval 1970–2007 as has been explained in Section 2.1; however, we admit that the reader needs reminding these aspects, and we have done so.*

- «The alongshore variations in transport» p. 16 line 397: Please be more specific mentioning those alongshore variations. Is this the small bump near Jurmala?

*We implicitly referred to Figure 6 that explains how alongshore variations in wave-driven transport are related with accumulation and erosion areas. We have made this reference explicit and have explained exactly what we mean in the revised version. We are not confident about what the small bump near Jurmala is showing. It may well be a numerical artifact that stems from the mismatch of the orientation of the shoreline and the N-S/E-W orientation of grid cells.*

- p. 17. Compute correlations over the years between identified cells would help to better characterize the link between those cells.

*Yes, this is helpful to understand what happens in long-term run in the area of frequent reversal and low net transport to the north of Ragaciems.*

- «long-term» p. 17 line 399; how is this riverine sediment flux taken into account in the model?

*This flux of sediment is ignored in the model. This is correct unless the flux is so intense that the added sediment changes the geometry of the shoreline within the study interval. We have added this remark to the text.*

- «sediment transport is high along the entire coastal stretch in years of intense transport (e.g., 1992) and low along the entire stretch in years of less intense transport (e.g., 1994)» p. 17 line 402: again, using statistical analysis over the whole period would make those conclusions drawn from 2 particular years much more robust.

*Thank you; we have added the relevant correlation coefficients and formal p-values.*

- «this property along the entire coastal segment» p. 17 line 404: not obvious after Jurmala

*Yes, we tell explicitly in the manuscript that "except in the immediate vicinity of the western breakwater of the Daugava River mouth". To make the description clearer, we have changed it to "except for about 6 km long stretch between the Lielupe River mouth and the western breakwater of the Daugava River".*

- More generally, the segments east and west of Jurmala seem to show distinct behaviors, which seems logical when considering the geographical position of Jurmala in figure 4.

*A small problem here is that Jurmala extends over 30 km along the seashore. It is customary to associate its formal location with the Majori district; however, different maps put "Jurmala" in quite different locations. In particular, in Fig. 4 the City of Jurmala extends from Kauguri to the letter "R" in Riga.*

*We agree that "the segments east and west of Jurmala seem to show distinct behaviors". However, as the focus of our study is different, we decided to not comment all different behaviours.*

- «While sediment from the easternmost system can be transported across the headland at Ragaciems, reverse transport is highly unlikely at an annual scale» p. 17 line 408 - 409: not clear can you explain how you deduce this from figure 9?

*Figure 9 shows that the net transport has a zero-upcrossing (and thus a clear divergence point) at this location in most of years (Fig. 9). The cited sentence explains this feature in other words. We have added a more rigorous formulation into the revised version.*

- «similar to the Latvian and Lithuanian Baltic proper shores, under a delicate balance of two predominant wave systems (Eelsalu et al., 2024b) that in this case work exactly again» p. 17 lines 422-423. Provide details.

*Yes, we have done so. As we have inserted cross-references to (new) section 2.2 already several times, we think it is sufficient to mention only the predominant wind and wave directions here.*

- «properties of waves generated by the north-north-western winds» p. 17 line 425: see comments above about wind and waves

*We believe that this location in the manuscript does not need any additional explanation as it should be fairly well known that a shorter fetch means smaller and shorter waves.*

- «Figure 10» p. 18. In figures 7,9,10, what is the meaning of no data?

*It has been explained above (Section 2.5) that clearly unrealistic or irrelevant estimates of transport are omitted.*

- « along the eastern shore of the Gulf of Riga » p. 18. For figures 7,9,10 that would be also useful to indicate the distance in km, in addition to points

*Yes and no. As the orientation of the coastline varies, different wave model grid points "cover" sectors of coastline with different length. The line composed of grid points follows a certain depth (because of the very nature of evaluation of wave-driven transport) and does not always match the geometry of the coast. This is to some extent visualised in Figure 5, for example, three grid points at the tip of Cape Kolka together represent a fairly small section of the shoreline. This feature makes it complicated to add km scale to these figures. Moreover, waves approaching from different directions may affect neighbouring sectors.*

- «The data for grid points that follow the orientation of breakwaters of the Port of Skulte, Salacgriva, Kuiviži, Treimani and at Kosmos are omitted.» p. 18. Same comment as for figure 7

*It is explained in Section 2.5 of the revised version that clearly unrealistic or irrelevant estimates of transport are omitted. We have inserted cross-reference to this explanation in these locations.*

- «of several man-made structures,» p. 19 line 449 : same comment as above how. How are these human made structures represented in the model at 600m resolution?

*It is explained in Section 2.5 of the revised version that the model in use does not adequately replicate the listed structures and that these structures stop most of wave-driven transport.*

- «waves from the northern directions dominate the wave-driven transport over this more than 30 km long» p. 19 line 466. Same comment as above about wind and waves

*It is our view that this formulation does not need any additional explanation: transport reversal means in this coastal stretch that transport goes to the south, and the possibility for this situation is that waves from the northern directions dominate the wave-driven transport.*

- «likely that wave-driven sediment flux passes this headland on many occasions and that the coastal segment from the Port of Skulte to Cape Kurmrags is a connected compartment.» p. 19 line 468-470: how do you support this statement?

*This conjecture stems from the observation that the location of the upcrossing associated with a minor headland near Lembuži varies by several kilometres in single years (Fig. 10). We have made this aspect explicit in the revised version.*

• «almost did not become evident on the shores of the Gulf of Riga where bulk potential transport even decreased to some extent 1990–2007 but net transport was at an almost constant level (Viška and Soomere, 2013a).» p. 20 lines 499-501. not very clear.
*Thank you, we have reformulated this and the previous sentence.*

• «A possible reason …» p. 20 line 502. Do the authors refer to the study of Soomere et al 2015 or to this study?
*We refer to Viška and Soomere (2013a) and have made this explicit in the revised version.*

• « of climate change » p. 20 line 503. How is it related to climate change? Wind? Could there be other factors?
*There has been extensive discussion of whether the feature identified in Soomere et al. (2015) is a reflection of climate change. This paper was published in journal called Climate Research. It is thus, at least to our understanding, fully appropriate to term this feature as "probable signal of climate change".*

• «As these shores are oriented very differently with respect to predominant wind directions, it is likely that such a signal is present on some of these shores only.» p. 20 lines 504-505. Same comment as above about wind and wave regimes and coastline configuration
*Thank you; we tell now that predominant wind directions are from the SW and (N)NW and that is highlighted in (new) Section 2.2.*

• «…cell is largest…» p. 20 line 505. Add "In the present study"
*This feature is evident both in historical estimates (Knaps et al., 1966; Ulsts, 1998) and earlier simulations; both represented in Fig. 2. We have only formulated a more rigorous result that was present but not made explicit in (Viška and Soomere, 2013) and refer the reader also to Fig. 2 in the revised version.*

• «shore and only about 30 % of that on the southern shore» p. 21 line 512. I guess you mean that transport on the southern shore is about 30% of the one on the western and even less of the eastern?
*No, we have in mind the ratio of transport on the southern/eastern shore and western/eastern shore. We have added the numerical values of transport to make the comparison explicit. The ratios are different for bulk and net transport.*

• «combination of the direction of predominant winds and orientation of the coastal segments» p. 21 lines 513-514. Same comment as above about wind
*Yes, we have added the predominant directions and a cross-reference to (new) Section 2.2 where the wind regime is described.*

• «does not increase gradually» p. 21 line 519 . It would help to compute the trend on figure 11: over the whole period and between 1990 and 2005 then 2005 and 2022
*We agree that providing the relevant numbers makes our arguments stronger. Shortly: average bulk transport per grid cell decreases in 1990–2005 in the entire study area and even more strongly on the eastern shore. This decrease is statistically significant at a 98% and 99.8% level, respectively. The similar decrease on the western and southern shore is much weaker and not statistically significant. The period 2005–2022 shows slow increase in bulk transport in all segments but this is far from being statistically significant. The particular numbers are accommodated in the caption to Figure 11 as they add very little to the main message of the paper.*

• «trend 2005–2022» p. 21 line 520. "Over 2005-2022" missing, here and in other places.
*The structure of English allows for omitting "over" or "in" in some such occasions as the dash is interpreted as 'to' in many version of English. The usage as 'trend 2005-2022 is common. Another approach would be to say 'in the period 2005-2022'. Anyway, we shall be happy to follow the journal style.*

• «along western, southern and eastern» p. 21 Figure 11. What are the limits of those segments?
*These segments are shortly depicted at the end of Introduction and described in detail in the last paragraph of (new) Section 2.3 and Fig. 4a.*

• «…from, or even in counterphase, with respect to…» p. 21 line 528. To be reformulated

*Yes, comma was in wrong place; however, we inserted brackets to make our point clear.*

•  «interannual variations 1990–2005 but has been almost steady since then» p. 22 line 534. This needs to be more quantitatively and statistically analysed (compute standard deviation, trends, etc...) and factors should be discussed (stormy years?)

*Thank you, the information of standard deviations in these years and shortly information about the weak trend in 2005–2022 is added into the revised version. The only driving factor are the wind patterns, and we mention that, with a reference to Eelsalu et al. (2024b) where these aspects have been addressed to some extent.*

•  «so-called storm season» p. 22 line 538. Differences and similarities with annual values should be discussed.

*Figs. 11 and 12 and their comments specifically address differences and similarities of annual and storm season values. We tell explicitly that "Additional information about the structure of the temporal course of transport is provided by analysis of transport during so-called storm seasons" after we have finished the description of annual values.*

•  «does not exhibit any significant trend» p. 22 line 542. Needs to be computed

*The relevant values have been added into the revised version to back up this claim.*

•  «course of this quantity signals that the decrease in annual values of the bulk transport along the western shore in the 1990s simply reflects improper clustering of the underlying data into annual values. » p. 22 line 543-544. Not clear.

*This minor comment was redundant and has been deleted.*

•  p. 22 Figure 12. I recommend to compute correlations between shores to see if they are statistically related

*Yes, we have done so and made explicit that correlation is strong and statistically significant for storm season values.*

•  «long-term and decadal change» p. 22 line 550: should be quantified

*We have added the trend slope; however, we left the claim about decadal changes only supported by the visual courses of net transport in Fig. 12.*

•  «the three coastal segments.» p. 23 line 552. Western and eastern shoes seem to be negatively correlated in figure 12?

*Yes, this is what we have explained in comments to Fig. 12 but used a different but even stronger wording: "Interestingly, most of these large variations are exactly in counter-phase on the western and eastern shores." We have provided more details in terms of correlation coefficients and statistical significance and reformulated the relevant sentences.*

•  «vital update ...» p. 23 line 558. Needs to be clearly explained: what are the new results here compared to previous studies

*Yes, this is exactly what we do in Sections 4.1, 4.2 and 4.3.*

•  «Figure 13» p. 24 Figure 13: erosion and accumulation areas should be highlighted.

*We admit that doing so might have some benefits for the reader; however, doing so seems unnecessary and potentially misleading as (1) a detailed map of such areas is available in Luijendijk et al. (2018, added to the list of references), (2) relationships between properties of potential transport and erosion and accumulation areas are not straightforward because of the assumption of unlimited availability of sediment, (3) our focus is quite different (identification of separated sedimentary compartments and cells) and inclusion of detailed discussion of erosion and accumulation would defocus the message and make the manuscript much longer.*

•  «mismatch of decadal variations in the wave-driven sediment transport in the interior of the Gulf of Riga (Viška and Soomere, 2013a) from those identified for longer segments of the eastern Baltic» p. 24 line 589-587. Not clear, should be more explicit

*We hope that a cross-reference to Section 3.4 where these aspects have been discussed in detail would help the reader. Also, the misleading reference (Viška and Soomere, 2013a) has been corrected to (Soomere et al., 2015).*

• «the specific orientation of some shore segments of the gulf with respect to the predominant moderate and strong wind» p. 24 line 594-595. Same comment, this needs an overview of wind regime

*Not here as the next lines provide a short but hopefully sufficient and clear explanation of the specific feature of the local winds: "predominant moderate and strong winds (usually south-western, and north-north-western, Soomere, 2003) that create the majority of waves responsible for sediment transport." The wind regime is described in "new" Section 2.2*

• «largely follows variations in…» p. 25 line 613. Same comment

*We hope that explanations in "new" Section 2.2 clarify the situation.*

• «mirrored pattern of time periods of high and low net potential transport on the western and eastern shores of the Gulf of Riga» p. 25 lines 618-619. Compute correlations to obtain more robust conclusions

*Yes, the correlations and associated p-values are added into the end of Section 3.4.*

• «almost regular fluctuations in the role of northerly winds in the system with almost constant amplitude and with a time scale of 3–4 years» p. 25 line 621 - 621. Not clear

*What we say should become clear from the appearance of Fig. 12: large fluctuations in net transport with a basically constant amplitude and a typical period of 3–4 years. As explained in previous paragraphs, it is mostly likely that they stem from variations in the northerly winds. We have however reformulated this conclusion as a possible conjecture as we have not analysed wind properties.*

On behalf of the co-authors
Tarmo Soomere, 02 January 2025

---

## Author Comment (AC3)

Dear Editor,
Dear Referee,
many thanks for these additional comments and suggestions. Even though some recommendations overlap with those provided by two existing reviews, quite many suggestions are definitely useful, and we are happy to amend the manuscript accordingly. Our response is provided using blue italics font.
Sincerely & on behalf of the co-authors
Tarmo Soomere

**Review of manuscript egusphere-2024-2640** Tarmo Soomere, Mikolaj Zbigniew Jankowski, Maris Eelsalu, Kevin Ellis Parnell, and Maija Viška: Alongshore sediment transport analysis for a semi-enclosed basin: a case study of the Gulf of Riga, the Baltic Sea

**1. General comments**

The authors study coastal sediment transport dynamics in the Gulf of Riga, a semi-enclosed bay at the eastern coast of the Baltic Sea using high-resolution wave time series (SWAN wave model) and the Coastal Engineering Research Centre (CERC) equations for the time span 1990-2022. Based on a hierarchical decomposition of the sedimentary near-coast sedimentary system into compartments and cells, the authors are able to specify transport dynamics along the coast. Integrating these local and regional features the authors draw a generalized picture of the wave-driven potential sediment transport dynamics of the study area. In addition to the natural meteorological, oceanographic and sedimentological environment also anthropogenic coastal structures influencing the transport processes are considered.

The work represents a valuable contribution to the model-based description of coastal dynamics of the eastern Baltic Sea. The methodology can be generalized to be applied to microtidal sandy coastal systems for sustainable coastal zone management. A publication is recommended after **moderate revision**.

*Thank you for this overall positive evaluation. We are happy adjust the manuscript to meet the comments and recommendations below.*

**2. Specific comments**

A restructuring of the text is suggested:

1. The **introduction** should primarily present the scientific task and the concept of its solution based on current knowledge, and quote to references (including authors' publications).
*This recommendation is to some extent opposite to what Referee #2 has suggested. We have make an attempt to meet at least partially all these suggestions by considerably expanding the Introduction in three directions: (i) presenting more detailed overview of the previous work on the subject and its results and limitations, (ii) moving a part of the material in Section 2.1 into Introduction as strongly suggested by Referee #2, (iii) making more clear what is new in our manuscript: the use of considerably increased spatial resolution that allows for the identification of features blocking sediment transport, update of the earlier estimates of wave-driven potential sediment transport rates, their interannual and decadal variations, location of divergence and convergence areas of the sediment flux and associated patterns of sedimentary compartments and cells on the sedimentary shores of the Gulf of Riga, plus understanding why long-term trends in sediment transport in the study area do not match similar trends on the Baltic proper shores.*

2. A next section should give an overview of the **study area** which is required especially for readers outside the Baltic region who are not familiar with the regional peculiarities. This concerns a description of the geographical, geological, climatic and oceanographic characteristics of the Baltic Sea before the

Gulf of Riga is described in more detail, whereby the sedimentological peculiarities of the coast should be taken into account. Sediment sources (including inputs from the open Baltic Sea and discharge from rivers) and sinks should be specified.

*This recommendation aligns with one of the main suggestions of Referee #2 who specifically recommended to provide a detailed overview of wind and wave climate and some aspects of climate change (e.g., the impact of the loss of sea ice) in the study area. We have done so by considerably expanding Section 2.2. The information about geographical characteristics of the study area is also expanded in Section 2.1 to cover all items that are necessary for adequate interpretation of the results. However, we are reluctant with respect of providing more information about the Baltic Sea as this information is not really needed for following our work and understanding what we have studied and what the meaning of our results is. We therefore hope that a reference to the general Baltic Sea textbooks (Feistel et al., 2005; Leppäranta and Myrberg, 2009) fills this gap. In a similar manner, the particular sedimentological peculiarities of the coast in the study area are not really decisive for our results and their interpretation as we only address potential sediment transport. Thus, here we also hope that a reference to an open access source where this description is given from the specific viewpoint of our studies (Viška and Soomere, 2013b) provides necessary resources. The issue of sediment flux from the Daugava River has been also raised by Referee #2. Additionally to arguments in the response to Referee #2, our apologies for not providing more data about sediment sources and sinks are that (i) our results are invariant with respect to sediment fluxes into and out of the system and (ii) we do not address sediment budget (that is important indeed but far out of the scope of our study).*

The **methods and data** section should include information on all primary (measured) or model-derived secondary data used, as well as a description of the models and their handling (such as the decomposition of the coastal space into cells and compartments and the model grid design). Regarding the models, it concerns the SWAN wave model and the ERA5 model for generating forcing data, the CERC equations and their parameterization.

*We fully agree that detailed description of the decomposition of the coastal space into cells and compartments and the model grid design is a vital part of the study and should be described in great detail. We have pretty much done so already in the original version, and have amended this description in Section 2.2 and made it more coherent also following recommendations of Referee #2.*

*We, however, are reluctant with respect of providing in this manuscript more detailed information about the wave model SWAN and its forcing ERA5. Both these items and their implementation has been described in numerous papers, including two our own papers about wave simulations in the study area (Giudici et al., 2023; Najafzadeh et al., 2024). Also, the ERA5 model and dataset are today a sort of standard and extremely well-described in the international literature data set. Thus, to our understanding, there is no need to include any more details about ERA5 into this particular manuscript additionally to the source (Hersbach et al., 2020).*

*We admit the some more information about how the simulations work (e.g., one-way nested system, the use of independent high-resolution grids) might be important to follow the line of thoughts, and we have included these aspects. Also, we have included reference to one more comparison of the use of ERA5 forcing near the study area.*

*The equations associated with the use of the CERC model and its particular implementation are described in detail in Section 2.5. We have expanded the description of how we interpret the outcome of this model in locations where the model may not work properly.*

In a separate section, the **results** already described in the present manuscript should be presented in a coherent manner. However, by now there is a discrepancy between the numerical model approaches and the purely qualitative verbal (or graphic) form of the result descriptions. This discrepancy could be

minimized through quantification (increased use of statistically estimated generalizing parameters and parameter functions).

*We agree that the original submission did not always separate new results from the pre-existing ones as also mentioned by other referees. We have made every attempt to make the presentation more coherent and highlight very clearly what is new. Additionally, as suggested also by Referee #2, we have added numerical values of the described statistical features (correlation coefficients, measures of statistical significance, slopes of trendlines, etc.) wherever we have reached any kind of conclusion that needed quantitative back-up.*

A separate **discussion** section is recommended. In this section, the acceptability of generalized data as model input should be discussed in particular.

*Here we are faced with opposing views of Referee #1 (who explicitly recommended to merge Results and Discussion sections) and Referee #3. The best we can do here is to provide the same argument as in the response to Referee #1, with hope that our position makes sense. Namely, we admit that we have, somewhat untraditionally, placed large parts of discussion directly after the relevant results. Our justification is twofold: (1) several results are counter-intuitive and probably needed some comments immediately, (2) properties of transport in different coastal stretches are greatly different, and we decided to help the reader by providing an immediate comparison wherever relevant and necessary.*

Other points are the reliability of the results and the limitations of the methods used. It is also important to refer here to the effects of anthropogenic structures in the coastal area on sediment dynamics, which are mentioned in various places in the text but are not yet discussed sufficiently.

*Yes, of course. We have considerably expanded Section 2.4 (2.5 in the revised version) towards explaining how we handle our results in locations where man-made structures may strongly distort sediment transport.*

In a final **summary and outlook** section, the results are to be concluded and a perspective is to be given. Figure 13 can be used as a basis for a graphic summary. However, the question arises whether the potential net transport could not be quantified by scaling the corresponding arrows.

*We have tried to do so but alongshore variations in the magnitude of sediment flux are so massive (many tens of times) that it was simply impossible to scale the arrows correspondingly. Thus, we have chosen to show at least qualitatively how sediment flux varies along the study area.*

In order to facilitate the understanding of the spatial and temporal relationships of the local model results, an additional tabular summary of the results is recommended.

*We definitely agree that quantitative estimates should be provided and we have done so in many locations following detailed recommendations of Referee #2. However, the number of meaningful/essential quantities is quite limited and we decided to provide them as part of body text in locations where they are relevant. To our eyes, doing so makes the text more transparent and easier to follow (compared to multiple references to various table entries).*

In the outlook, the spatial extension of the investigations already indicated in the last sentence of the manuscript, as well as a methodological refinement for sustainable coastal zone management, should *be addressed in more detail.*

*We have expanded this discussion to cover several other potential developments of the model, such as the use of variable locations of nearshore wave model grid cells to better replicate wave properties at the breaker line.*